# Mind Your Augmentation: The Key to Decoupling Dense Self-Supervised Learning

**Congpei Qiu**[1*] **Tong Zhang**[2*] **Yanhao Wu**[1] **Wei Ke**[1‡] **Mathieu Salzmann**[2] **Sabine Süsstrunk**[2]

[1]School of Software Engineering, Xi'an Jiaotong University, China
[2]School of Computer and Communication Sciences, EPFL, Switzerland

## Abstract

Dense Self-Supervised Learning (SSL) creates positive pairs by building positive paired regions or points, thereby aiming to preserve local features, for example of individual objects. However, existing approaches tend to couple objects by leaking information from the neighboring contextual regions when the pairs have a limited overlap. In this paper, we first quantitatively identify and confirm the existence of such a coupling phenomenon. We then address it by developing a remarkably simple yet highly effective solution comprising a novel augmentation method, Region Collaborative Cutout (RCC), and a corresponding decoupling branch. Importantly, our design is versatile and can be seamlessly integrated into existing SSL frameworks, whether based on Convolutional Neural Networks (CNNs) or Vision Transformers (ViTs). We conduct extensive experiments, incorporating our solution into two CNN-based and two ViT-based methods, with results confirming the effectiveness of our approach. Moreover, we provide empirical evidence that our method significantly contributes to the disentanglement of feature representations among objects, both in quantitative and qualitative terms.

## 1 Introduction

Self-supervised learning (SSL) has achieved significant progress, now surpassing traditional supervised pre-training when applied to diverse domains and tasks (Chaitanya et al., 2020; Ericsson et al., 2021; Caron et al., 2021; Wu et al., 2023). SSL methods aim to maximize the similarity between the feature representations of positive pairs. However, positive pairs are built with strong augmentation and their features are obtained by average pooling for each view, leading to inconsistent semantic alignment (Zhang et al., 2022) and a lack of localization information. Consequently, this approach is most suitable for well-curated datasets and becomes problematic when applied to images containing multiple objects or tasks that necessitate precise location information. To alleviate the constraints imposed by SSL, recent literature has witnessed substantial efforts in the domain of dense-level SSL. At the core of these efforts is the idea of defining positive pairs by identifying matching locations, thereby preserving distinct semantic cues.

In the context of dense SSL, cutout and masking are pivotal components for both CNNs and ViTs, as they contribute to the creation of meaningful positive pairs. Point-level CNN-based methods (Wang et al., 2021; Xie et al., 2021) construct positive pairs by selecting matched points in feature space, while others, including both region-level CNN-based (Wei et al., 2021) and ViT-based methods (Zhou et al., 2022; Xue et al., 2023), rely on image coordinates as oracle and employ various augmentations such as cutout and masks to generate positive pairs. These pairs are subsequently utilized in teacher-student networks for feature alignment. However, a challenge arises when there is limited overlap between the views, often due to mismatches or over-cuts. In such cases, the features extracted from the query view tend to take shortcuts by leveraging information from the background to minimize the self-supervised learning (SSL) loss. This unintentional mechanism leads to closer proximity of features from unrelated regions, ultimately resulting in less discriminative regions being dominated by their neighbors. We term this phenomenon *coupling*, as depicted in Fig. 1.

In this paper, we present a simple yet highly effective strategy that seamlessly integrates into recent dense SSL methods to address the object coupling issue. We observe that existing augmentation

---

* These authors contributed equally to this work.    ‡ Corresponding author
Code is available at https://github.com/ztt1024/denseSSL

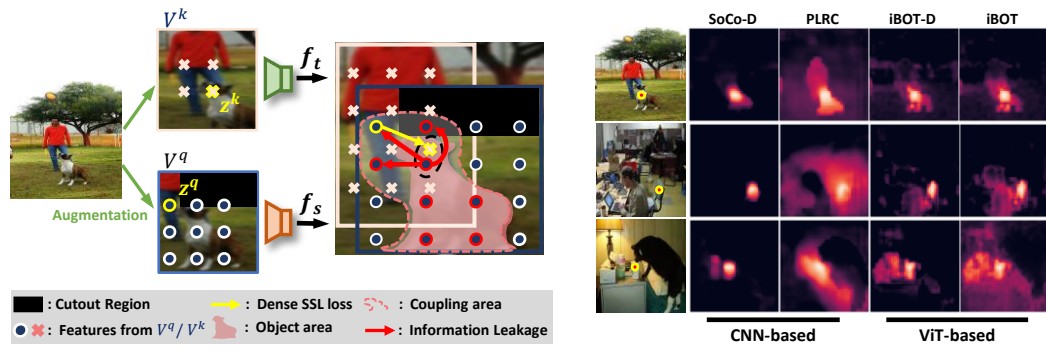

(a) Illustration of coupling    (b) Point affinity visualizations

Figure 1: **Visualization of Coupling Effects.** (a) When positive pairs are mismatched or share small overlapping unmasked regions, the features in the query $z^q$ are pushed towards the key features $z^k$, causing the key feature's semantics to propagate to a wider query semantic area (less discriminative). (b) The visualization depicts affinity maps generated by different methods, with each map showing feature similarity between a single point (marked by a red dot) and the rest of the image. '-D' denotes a model combined with our decoupling strategy.

techniques, such as blockwise masks (Zhou et al., 2022) and cutout (DeVries & Taylor, 2017), apply masks or cutouts independently to the input view, without taking into account the presence of other masked or cut regions. This prevents the student views from providing sufficient semantic cues to align with the teacher encoders, especially when substantial occlusions occur, e.g., due to the overlap of multiple masked or cut regions. To address this challenge, we introduce Region Collaborative Cutout (RCC), a method designed to avoid creating excessively large masked areas that could lead to information leakage from distant regions. Our approach involves creating regions uniformly and implementing cutouts within these regions, starting with larger ones and progressively moving to smaller areas. The cutout operations among regions work collaboratively to ensure that no large masks exceed a specified threshold. More importantly, we introduce a de-coupling branch to eliminate incorrect dependencies between each region and its surroundings by adding contextual noise to the surroundings of sampled regions within images. Specifically, we construct de-coupled views by replacing the background regions of key views defined by RCC masks with randomly sampled background images. We then directly apply the self-supervised learning loss to the features extracted from the same foreground region in both the decoupled and key views. This design allows the network to learn independently from each local region, promoting the capture of robust representations less influenced by irrelevant context. Importantly, our de-coupling branch is compatible with existing methods and requires only a modification of the augmentation pipeline.

In a nutshell, our contributions can be outlined as follows:

1. We tackle the challenge of coupling in dense SSL, offering quantitative empirical evidence to validate its presence across different methods with both CNNs and ViTs.

2. We introduce a novel augmentation method tailored for dense SSL, named *Region Collaborative Cutout (RCC)*. This method plays a pivotal role in reducing object coupling by effectively reducing the difficulty of aligning features in masked regions.

3. Building upon the foundation of our new augmentation method, we further enhance feature robustness to noise by introducing a decoupling branch. This branch ensures that features are learned independently from their surroundings.

We showcase our contributions through comprehensive experiments and ablation studies, providing empirical validation for the effectiveness of our entire method as well as its individual components, namely RCC and the decoupling strategy.

## 2    RELATED WORK

**Self-supervised Learning.** With the introduction of InfoNCE (Oord et al., 2018), inspired by the triplet loss in metric learning (Oh Song et al., 2016; Sohn, 2016), contrastive learning has become

the dominant SSL strategy. SimCLR (Chen et al., 2020a) constructs positive pairs by applying random data augmentations twice to the same image and treats other images as negatives. MoCo (He et al., 2020) explores simpler architecture, and BYOL (Grill et al., 2020) starts the trend of simply minimizing positive pairs' distance. Other explorations include clustering-based (Asano et al., 2019; Caron et al., 2018; 2020; Ji et al., 2017; Zhang et al., 2019b;a) and feature decorrelation (Zbontar et al., 2021). These methods primarily focus on capturing semantic invariance through *image-level* augmentation. Consequently, they are less suitable for images with multiple objects (He et al., 2019; Zhang et al., 2022) and dense prediction tasks, where local context becomes crucial.

**Dense Self-supervised Learning.** Dense SSL methods aim to enhance the representation of regions by constructing positive pairs with finer granularity. Notably, the point-level methods, including DenseCL (Wang et al., 2021), PixPro (Xie et al., 2021), and VADeR (O Pinheiro et al., 2020), establish positive point pairs by evaluating pairwise cosine distances among all points from two augmented views and selecting those with higher similarity. However, these methods focus solely on point similarity and do not consider the relationships with their neighbors, leading to inaccurate pairings. In response, SetSim (Wang et al., 2022b) and Point-level RCL (Bai et al., 2022) seek to refine the selection of positive pairs by taking neighboring points into account. Nonetheless, their strategies heavily rely on image-level SSL, resulting in mismatches in the selection of positive pairs.

On the other hand, region-level methods focus on identifying matching regions and applying SSL loss to features extracted from these regions. Self-EMD (Liu et al., 2020) minimizes the Earth mover's distance between two paired regions; InsLoc (Yang et al., 2021) utilizes pretext tasks to facilitate region-level contrastive learning; DetCon (Hénaff et al., 2021) and SoCo (Wei et al., 2021) employ unsupervised masks and bounding boxes, respectively, as external labor-free supervision to build positive pairs. These region-based networks share a similarity with image-level SSL, where only the most discriminative parts can be learned due to average pooling. In the realm of ViT structures, recent works use masked image modeling (MIM) as augmentations to reconstruct masked patches in color space (Xie et al., 2022; He et al., 2022) or the features of masked patches (Bao et al., 2022; Xue et al., 2023). Among these approaches, iBOT (Zhou et al., 2022) serves as the core component of the vision foundation model DINOv2 (Oquab et al., 2023). Despite their impressive results, prior Dense SSL methods have overlooked the issue of feature coupling.

**Augmentations.** Most of the augmentations, e.g., cutout (DeVries & Taylor, 2017), gaussian blur, and random crops, are designed for supervised image classification tasks. Some alternatives, including mixup (Zhang et al., 2017), cutmix (Yun et al., 2019), and TokenMix(Liu et al., 2022), are also effective for supervised semantic segmentation and object detection. SimCLR (Chen et al., 2020b) was the first to adopt augmentation in SSL, evidencing that SSL benefits from stronger augmentation. Recently, SDMP (Ren et al., 2022) integrates the mixup method into SSL frameworks. Additionally, patch-based augmentations (Qin et al., 2022) (rotations, infill, and shuffle) have been proposed for ViT's pre-training robustness against noise. However, these augmentation methods are tailored to enhance image-level discriminative features for classification tasks, often overlooking the essential localization information due to its inherently complex nature.

While dense SSL has emerged as a promising direction, particularly replacing image-level approaches, many challenges associated with feature coupling in Dense SSL remain unaddressed, particularly in scenarios involving images with multiple objects. We aim to tackle this problem by introducing novel augmentations and network modules specifically designed for Dense SSL.

## 3 MOTIVATION: EMPIRICAL STUDY

We initiate our work with an empirical analysis that aims to shed quantitative light on the issue of coupling in dense SSL. Let $\boldsymbol{X} \in \mathbb{R}^{H \times W \times 3}$ denote a sample image from a pre-training dataset $\mathbb{D}$, $\mathcal{P}$ is the region where the feature is extracted from, and $\mathcal{P}^C$ is the surrounding region providing contextual information. Note that each region may contain either a single or multiple patches. Thus, the set of image patches corresponding to $\mathcal{P}$ is represented by $\boldsymbol{X}_{\mathcal{P}}$. With an encoder $E : \mathbb{R}^{H \times W \times 3} \to \mathbb{R}^{h \times w \times c}$, the feature representation of $\mathcal{P}$ is computed as $f(\boldsymbol{X}, \mathcal{P}) = g(E(\boldsymbol{X}), \mathcal{P})$, where $g$ denotes feature extraction from regions. For an augmented query view $\boldsymbol{X}^q$ and a key view $\boldsymbol{X}^k$, we formulate dense SSL as

$$\mathcal{L}_{\text{Dense}}(\boldsymbol{z}_{\mathcal{P}_q}^q, \boldsymbol{z}_{\mathcal{P}_k}^k) = \mathcal{L}_{SSL}\left[f_s(\boldsymbol{X}^q, \mathcal{P}_q), f_t(\boldsymbol{X}^k, \mathcal{P}_k)\right], \tag{1}$$

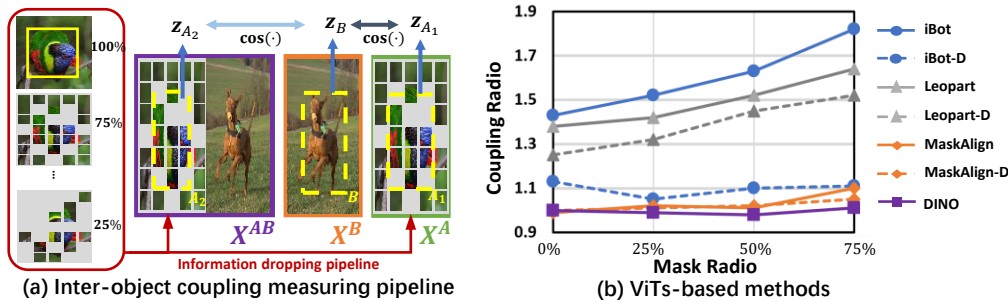

Figure 2: **Measuring inter-object coupling in ViTs.** (a) Random masking is applied to image $\boldsymbol{X}^A$ with varying ratios, combined with an image $\boldsymbol{X}^B$ to create image $\boldsymbol{X}^{AB}$. Yellow boxes highlight regions from which features are extracted. (b) We report the $CR$ across different mask ratios for ViT-based models. The -D indicates the method integrated with our decoupling strategy.

where $f_t$, $f_s$ denote the teacher and student branch. $\mathcal{L}_{SSL}$ is any existing SSL loss such as InfoNCE and $\ell_2$. The primary objective of this framework is to facilitate the learning of semantically invariant features, ensuring that the features extracted from region $\mathcal{P}_q$ closely correspond to those from $\mathcal{P}_k$. However, this objective also necessitates the retrieval of semantic information from the context $\mathcal{P}^C$ when region $\mathcal{P}_q$ lacks sufficient semantic cues to match region $\mathcal{P}_k$.

Our intuition for measuring coupling is to examine the level of independence exhibited by local features when exposed to varying contexts. To better investigate the coupling issue, we showcase coupling in both inter-object and intra-object scenarios. Note that image-level SSL is obtained by replacing the region extractor with average pooling over the entire input view. In this case, there is no risk of information leakage through spatial shifts.

## 3.1   INTER-OBJECT COUPLING

We first investigate the inter-object case as depicted in Fig. 2(a), where we consider pairs of objects ($A$ and $B$) belonging to different categories. Specifically, we construct three views: $\boldsymbol{X}^{AB}$ containing both $A$ and $B$, and $\boldsymbol{X}^A$ and $\boldsymbol{X}^B$ containing an individual object. To evaluate the extent of entanglement between two objects, we randomly remove patches of object $A$ with different ratios in the image $\boldsymbol{X}^{AB}$, and extract their features through backbones pre-trained with different methods:

$$\boldsymbol{z}_{A_1} = f(\boldsymbol{X}^A, \mathcal{P}_{A_1}), \; \boldsymbol{z}_{A_2} = f(\boldsymbol{X}^{AB}, \mathcal{P}_{A_2}), \; \boldsymbol{z}_B = f(\boldsymbol{X}^B, \mathcal{P}_B), \tag{2}$$

where $\mathcal{P}_{A_1}, \mathcal{P}_{A_2}, \mathcal{P}_B$ are the regions containing $A_1$, $A_2$ and $B$. We then define the coupling ratio as

$$CR = \frac{\max\left(\frac{\pi}{2} - \theta(\boldsymbol{z}_{A_2}, \boldsymbol{z}_B), \epsilon\right)}{\max\left(\frac{\pi}{2} - \theta(\boldsymbol{z}_{A_1}, \boldsymbol{z}_B), \epsilon\right)}, \tag{3}$$

where $\theta(\cdot, \cdot)$ is the angle between two feature vectors, and $\epsilon$ is a scalar introduced for numerical stability. $CR > 1$ indicates information leakage from neighbor $\boldsymbol{X}^{AB}_{\mathcal{P}_B}$ to $\boldsymbol{X}^{AB}_{\mathcal{P}_A}$. Our findings are presented in Figure 2(b), which illustrates the results obtained from different pre-trained models. Specifically, when region $A$ loses more semantic information, the feature $\boldsymbol{z}_{A_2}$ tends to converge towards $\boldsymbol{z}_B$, providing clear evidence of the existence of inter-object shortcuts. As expected, in the case of image-level methods, $CR$ tends to approach 1, as no context regions are involved during the pre-training stage. The same trends can be found in CNNs, and more details can be found in Appendix B.

## 3.2   INTRA-OBJECT COUPLING

Apart from the entanglement between objects, coupling within individual objects is also prevalent. We begin by sampling a ground-truth object mask denoted as $\mathcal{M}$ along with its image $\boldsymbol{X}$. From this mask, we select a quarter to represent the object part, denoted as $\mathcal{M}op$. We then remove the remaining portions of the object mask, resulting in a modified image $\boldsymbol{X}^{OP}$, and construct $\boldsymbol{X}^C$ by replacing the rest parts with a randomly sampled background, as shown in Fig 3(a). To extract region-level features of the object part, we perform masked average pooling using $\mathcal{M}_{op}$ on the

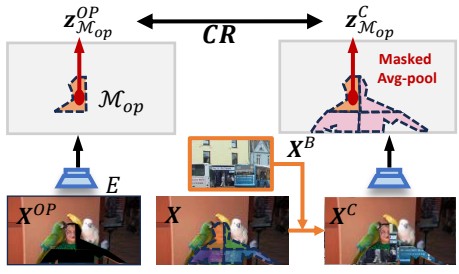 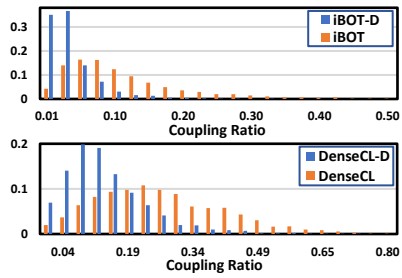

(a) Intra-object coupling measuring pipeline      (b) Histogram of coupling ratio

Figure 3: **Measuring intra-object coupling.** (a) We select a quarter of the object $\mathcal{M}_{op}$ as object part. The views $\boldsymbol{X}^{OP}, \boldsymbol{X}^{C}$ are constructed by masking out the remaining object area and replacing it with sampled background. The $CR$ is calculated between region-level features extracted using masked average pooling. (b) We draw the $CR$ histograms of the pre-trained models.

feature map obtained by forwarding the constructed view through the backbone network $E$. We define the coupling ratio within objects as:

$$CR = 1 - \cos\left(\boldsymbol{z}_{\mathcal{M}_{op}}^{OP}, \boldsymbol{z}_{\mathcal{M}_{op}}^{C}\right). \tag{4}$$

A higher value of $CR$ indicates that the features of the object part are more inclined to the influence of surroundings. Figure 3(b) displays histograms of the computed coupling ratios for the sampled object parts, underscoring how irrelevant background significantly affects representations of object parts. The mean of intra-object CR clearly decreases with our strategy.

As demonstrated by the above experiments, the region in a lack of semantics due to stronger masking tends more to retrieve information from the context. Consequently, we further verify that the coupling will gradually increase as the augmented views become more challenging in Appendix C.1.

## 4 DE-COUPLING FROM DENSE-LEVEL SHORTCUTS

Our empirical studies have brought to light the existence of dense-level shortcuts that impede the models from effectively capturing local visual semantics. In this section, we introduce our de-coupling strategy as a solution to mitigate these challenges. Importantly, our de-coupling strategy is designed for general applicability, and we will show how to integrate our method into existing dense-level pre-training methods.

### 4.1 DE-COUPLING STRATEGY

As discussed in Section 3, strong augmentations with higher masking ratios force the student networks to search for shortcuts to align the features with those from the teacher branch. To counter this, our strategy starts with proposing a new augmentation method, namely Region Collaborative Cutout (RCC), to generate views with reduced difficulty levels to block the shortcut to contextual information. Moreover, we introduce a mixture of cropped views with various backgrounds to mitigate the bias from the contextual information of a single image.

#### 4.1.1 DECOUPLED VIEWS GENERATION

To create the decoupled view, we leverage RCC to generate masks for extracting foreground information from the key view while filling the unmasked region with a randomly sampled image as background context information. Without loss of generality, we first divide the input into $N \times N$ grids and randomly create a single bounding box with a pre-determined range of scale and aspect ratio within each grid as regions, yielding $\{\boldsymbol{b}_i\}_{i=1}^{N \times N}$ regions with diverse scales and positioned uniformly across the image. Each box is then used as anchor to collaboratively perform cutout to obtain the mask $\mathcal{M}_{RCC}$. Unlike blockwise mask (Zhou et al., 2022) and cutout on proposal (Wei et al., 2021), where undesirable large masks or cuts can be attained, RCC initiates mask generation in a descending order based on the bounding box region size, simultaneously addressing overcut regions caused by larger ones. For the $i$-th step, we calculate the actual cutout ratio $r_{ci}$ incurred by the previous steps. If it is higher than pre-defined $r_c$, we will restore the region and perform a new random

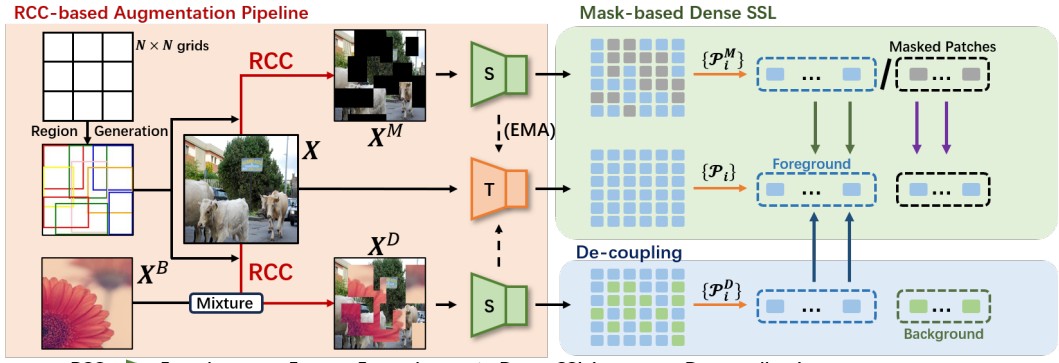

Figure 4: **Structure of RCC and decoupling strategy.** Our RCC is applied in the augmentation stage to replace previous masking methods. Given an input image $\boldsymbol{X}$ with generated proposal regions, we replace the original mask strategy with our RCC, leading to masked view $\boldsymbol{X}^M$ for student encoder. We construct the de-coupled view $\boldsymbol{X}^D$ with $\boldsymbol{X}$ and randomly sampled $\boldsymbol{X}^B$ as an additional student branch. $\{\mathcal{P}_i\}$ denotes the regions that we used to constrcut positive pairs, which can be either single patches or regions. Note that, the decoupling does not reconstruct the background in de-coupling branch.

cutour with ratio $r_c$. Otherwise, another cutout region with ratio $r_c - r_{ci}$ will be applied. We defer the pseudo-code in Alg. 2

The $\mathcal{M}_{RCC}$ masks provide effective augmentation for dense SSL when pre-trained on multi-object datasets. Fig. 5 shows the difference between the Cutout and our RCC. Our method prevents masks of different regions from being connected in the same place, which blocks the semantic information for the student encoder. Comparison with blockwise masking and additional results can be found in Appendix D.

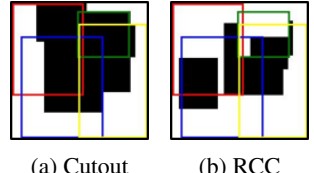

(a) Cutout     (b) RCC

Figure 5: **Cutout vs. RCC.** Masks are generated for each bounding box.

We then generate the de-coupled view $\boldsymbol{X}^D$ of an image $\boldsymbol{X}$ with a randomly sampled background image $\boldsymbol{X}^B$ and the binary matrix $\boldsymbol{M}_{RCC}$ of $\mathcal{M}_{RCC}$:

$$\boldsymbol{X}^D = \boldsymbol{X} \odot (\boldsymbol{I} - \boldsymbol{M}_{RCC}) + \boldsymbol{X}^B \odot \boldsymbol{M}_{RCC}. \tag{5}$$

### 4.1.2 DE-COUPLING LOSS

In the de-coupled view $\boldsymbol{X}^D$, each region is surrounded by the remaining unmasked context and context noise controlled by $\mathcal{M}_{RCC}$. For a feature extraction region $\mathcal{P}$ that has overlap with unmasked region $\mathcal{M}_{RCC}^C$, we directly align the dense-level feature defined by $\mathcal{P} \cap \mathcal{M}_{RCC}^C$ to the corresponding one on original view $\boldsymbol{X}$. This can be expressed as

$$\mathcal{L}_{\text{DC}} = \mathcal{L}_{SSL}(\boldsymbol{z}_{\mathcal{P}}^D, \boldsymbol{z}_{\mathcal{P}}^F) \text{, with } \boldsymbol{z}_{\mathcal{P}}^D = f_s(\boldsymbol{X}^D, \mathcal{P} \cap \mathcal{M}_{RCC}^C), \ \boldsymbol{z}_{\mathcal{P}}^F = f_t(\boldsymbol{X}, \mathcal{P}). \tag{6}$$

Note that $\mathcal{P}$ can be a bounding box, a single patch token in ViTs, or a point in CNNs. Then, we apply the SSL loss to each $\mathcal{P}$ to achieve de-coupling. With the above objective, the dense-level feature $\boldsymbol{z}_{\mathcal{P}}^D$ is tasked with distinguishing between incomplete yet relevant context and introduced noise while predicting $\boldsymbol{z}_{\mathcal{P}}^F$ with a global context. This both de-couples the model from entangled irrelevant contexts and enhances the out-of-context learning capacity.

### 4.2 COMBINATION WITH DENSE-LEVEL SSL

Our decoupling strategy seamlessly integrates into existing dense-level frameworks, as illustrated in Fig. 4. In addition to the query view $\boldsymbol{X}^q$ and key view $\boldsymbol{X}^k$, we generate a decoupled view $\boldsymbol{X}^D$ using the key view $\boldsymbol{X}^k$ and masks $\mathcal{M}_{RCC}$ generated by RCC. It's important to note that the background image $\boldsymbol{X}^B$ is randomly selected from each batch. Given the set of feature extraction positions $\mathcal{P}_i$ as positive pairs, the dense SSL objective with decoupling is expressed as:

$$\mathcal{L}_{Dense-D}(\boldsymbol{X}^q, \boldsymbol{X}^k, \boldsymbol{X}^D) = \frac{1 - \lambda_{DC}}{|\{\mathcal{P}_i\}|} \sum_{\mathcal{P}_i} \mathcal{L}_{Dense}(z_{\mathcal{P}_i}^q, z_{\mathcal{P}_i}^k) + \frac{\lambda_{DC}}{|\mathbb{P}_D|} \sum_{\mathcal{P}_j^k \in \mathbb{P}_D} \mathcal{L}_{DC}(z_{\mathcal{P}_j}^D, z_{\mathcal{P}_j^k}^k), \tag{7}$$

where $\lambda_{DC}$ represents the de-coupling weight, $\mathbb{P}_D$ is defined as the set of feature extraction positions that are covered by $\mathcal{P}_j^k$ and intersect with the foreground $\mathcal{M}_{RCC}^C$. The notation $|\cdot|$ denotes the cardinality of the set, which can correspond to either the number of patches (in the context of point-level methods) or regions (for region-level methods). $\mathcal{L}_{DC}$ applies the same type of loss as $\mathcal{L}_{Dense}$ and thus requires no further modification. More details and examples of combining our strategy with other dense SSL frameworks are provided in Appendix E.1.

## 5 EXPERIMENTS

### 5.1 IMPLEMENTATION DETAILS

**De-coupling Setting.** To demonstrate the effectiveness and generalization capability of our de-coupling strategy, we apply our module on DenseCL (Wang et al., 2021), SoCo (Wei et al., 2021), Leopart (Ziegler & Asano, 2022), iBOT (Zhou et al., 2022) and MaskAlign (Xue et al., 2023), which include both CNN-based and ViT-based methods. With the exception of SoCo, which utilizes Selective Search to generate proposals as region prior, we apply $3 \times 3$ grids to randomly generate regions for the other methods. For the backbone, we employ ResNet-50 (He et al., 2016) on CNN-based methods and ViT-S/16 (Dosovitskiy et al., 2021) on Vit-based methods. We defer the evaluation of Leopart to Appendix F as it aims for unsupervised segmentation and has different evaluation metrics.

**Pre-training Setting.** In the pre-training stage, we sample only $25\%$ images from a batch to construct the de-coupling branch. Note that the de-coupled views will only be forwarded to the student encoder, with acceptable additional computational costs (reported in Appendix F). As we target SSL on the multi-object datasets, following the protocol of (Bai et al., 2022; Wang et al., 2021), we pre-train each model on COCO *train2017* for 800 epochs. For a fair comparison, we adopt the same hyper-parameter for every method with or without the de-coupling strategy.

### 5.2 DENSE PREDICTION

**Evaluation Protocols.** For CNN-based methods, we follow the dense prediction protocols of MoCo-v2. On VOC (Everingham et al., 2010) detection tasks, we fine-tune a Faster R-CNN structure Ren et al. (2015), whereas on COCO (Lin et al., 2014) detection and instance segmentation, we adopt a Mask R-CNN (He et al., 2017) structure. For ViT-based methods, we follow the same protocol as iBOT, where we apply Cascade Mask R-CNN (Cai & Vasconcelos, 2019) for object detection and instance segmentation on COCO. For semantic segmentation on ADE20K (Zhou et al., 2017), we employ the task layer in UperNet (Xiao et al., 2018) and fine-tune the whole network.

Table 1: **Main results pre-training with CNN-based methods.** '-D' denotes combining with de-coupling strategy. We pre-train models marked by † with their official codes and setting on COCO.

| Method | VOC Det. | | | COCO Det. | | | COCO ISeg. | | |
|--------|------|-----------|-----------|------|-----------|-----------|------|-----------|-----------|
| | AP | $AP_{50}$ | $AP_{75}$ | AP | $AP_{50}$ | $AP_{75}$ | AP | $AP_{50}$ | $AP_{75}$ |
| MoCo v2† | 54.6 | 81.0 | 60.4 | 37.8 | 57.4 | 41.0 | 32.9 | 54.1 | 35.2 |
| ReSim† | 56.6 | 81.7 | 63.5 | 38.3 | 57.8 | 41.4 | 33.5 | 54.4 | 35.6 |
| DenseCL | 56.7 | 81.7 | 63.0 | 38.5 | 58.1 | 41.5 | 33.6 | 54.8 | 35.7 |
| DenseCL-D | 57.2 | 82.2 | 63.7 | 39.3 | 58.7 | 42.6 | 34.2 | 55.7 | 36.5 |
| PLRC | 57.1 | 82.1 | 63.8 | 39.8 | 59.6 | 43.7 | **35.9** | 56.9 | **38.6** |
| SoCo† | 56.8 | 81.7 | 63.5 | 38.5 | 57.9 | 41.5 | 33.4 | 54.6 | 35.4 |
| SoCo-D | **57.8** | **82.5** | **64.4** | **40.3** | **60.1** | **44.0** | 35.1 | 56.9 | 37.6 |

**Results of CNN-based Methods.** Tab. 1 reports the performance on CNN-based methods. We compare three types of methods according to feature extraction strategy: 1) Image-level, *i.e.*, directly perform average pooling on the whole feature map, like MoCo v2 (Chen et al., 2020b); 2) Point-level, *i.e.*, every feature vector as positive pairs are used for SSL, including DenseCL (Wang et al., 2021) and PLRC (Bai et al., 2022); 3) Region-level, which uses a set of features from the feature map followed by average pooling, such as ReSim (Xiao et al., 2021) and SoCo (Wei et al., 2021). Considering the substantial memory requirements of PLRC, we demonstrate the integration of our

method into DenseCL as a representative point-level approach. Additionally, we select SoCo as a representative region-level method for evaluation. It's important to note that all dense SSL methods consistently outperform image-level SSL methods in downstream tasks. Particularly noteworthy is the impressive performance improvement achieved by both DenseCL and SoCo when incorporating our de-coupling strategy, surpassing their original results.

**Results of ViT-based Methods.** In Tab.2, we present a comparison between dense SSL approaches utilizing ViTs, both with and without our de-coupling strategy. The results show significant improvements when employing our strategy. Specifically, iBOT with our de-coupling achieves significant performance gains, surpassing its original baseline by a large margin of 2.8 AP, 2.1 AP, and 1.7 mIoU across three tasks. Similarly, MaskAlign-D also demonstrates remarkable performance enhancements compared to its original version. These comparisons underscore the efficacy of our de-coupling strategy in aiding dense SSL methods to more effectively capture dense semantics while mitigating the issues associated with coupling. Importantly, our approach exhibits versatility, as it can be readily applied to various SSL strategies and backbone architectures.

Table 2: **Main results pre-training with ViT-based methods.** '-D' denotes combining with de-coupling strategy.

| Method | COCO Det. | | | COCO ISeg. | | | ADE Seg. |
| --- | --- | --- | --- | --- | --- | --- | --- |
| | AP | $AP_{50}$ | $AP_{75}$ | AP | $AP_{50}$ | $AP_{75}$ | mIoU |
| iBOT | 42.3 | 61.2 | 45.6 | 37.0 | 58.3 | 39.4 | 39.9 |
| iBOT-D | **45.1** | **64.3** | **48.7** | **39.1** | **61.2** | **41.7** | **41.6** |
| MaskAlign | 45.6 | 65.2 | 49.7 | 39.6 | 62.0 | 42.4 | 43.7 |
| MaskAlign-D | **46.7** | **66.4** | **50.5** | **40.5** | **63.2** | **43.5** | **44.3** |

## 5.3 OBJECT-LEVEL KNN

We appreciate the simplicity and effectiveness of $k$-NN, which directly reflect the quality of learned image-level representations, so we extend it to dense-level tasks and obtain object-level $k$-NN (OKNN). Concretely, we extract object-level features from the final feature map with ground-truth bounding boxes using RoIAlign and average pooling. We extract $N$ object features per image from the training set along with corresponding labels. Similar to image-level $k$-NN, we predict the label for each object in the evaluation set by finding the $k$-nearest object-level features in the training set.

**Disturbed OKNN.** To quantify the influence of coupling for multi-object images, we further conduct disturbed OKNN (OKNN-D) by replacing the region of each image uncovered by $N$ selected bounding boxes with randomly sampled background images. The feature extraction and prediction protocols are kept the same.

**Comparison.** Tab. 3 shows the O-KNN and OKNN-D accuracy on COCO using *train2017* for feature extraction and *val2017* for evaluation. The proposed decoupling strategy boosts the performance of both OKNN and OKNN-D when combined with other dense SSL methods. The improvement on OKNN-D is more obvious, demonstrating the effectiveness of our approach to improving model robustness for irrelevant contexts.

Table 3: **Main Results of OKNN and OKNN-D.** We evaluate models on COCO and report the top-1, top-5 accuracy.

| Method | OKNN | | OKNN-D | |
| --- | --- | --- | --- | --- |
| | *Top*.1 | *Top*.5 | *Top*.1 | *Top*.5 |
| DenseCL | 72.5 | 89.6 | 41.5 | 59.9 |
| DenseCL-D | **74.5** | **91.2** | **45.1** | **63.9** |
| SoCo | 73.5 | 89.7 | 43.7 | 61.2 |
| SoCo–D | **75.8** | **90.8** | **47.8** | **65.0** |
| iBOT | 75.2 | 90.8 | 45.8 | 68.3 |
| iBOT-D | **78.7** | **93.3** | **52.1** | **71.9** |
| MaskAlign | 77.4 | 91.6 | 53.8 | 74.0 |
| MaskAlign-D | **79.0** | **92.7** | **55.2** | **76.1** |

## 6 ABLATION STUDY

We dissect our de-coupling strategy and study the impact of each to reveal the strengths of our designs here. In ablation studies, we pre-trained iBOT and DenseCL on COCO for 200 epochs. More details and analysis of our RCCand ablation results of DensCL are in Appendix D and G.

## 6.1 MASK-TYPE FOR DE-COUPLING

To demonstrate the effectiveness of our RCC, we conducted experiments utilizing various types of masking for de-coupling, including: a) $DC_{Rec}$: Utilizing a random rectangular region as the foreground; b) $DC_{Ran}$: Randomly mask out patches; c) $DC_{Blc}$: Employing blockwise masks like BEiT (Bao et al., 2022); d) $DC_{Cutout}$: Replacing our RCC with Cutout. In Tab. 4, we report the results, including the Top-1 accuracy of OKNN, AP in COCO detection, and the average coupling ratio for both inter-object and intra-object levels, as

Table 4: **Comparison of mask types for de-coupling.** iBOT is used as a baseline, we only change the masking strategy for de-coupling.

| Method | COCO Det. | OKNN | $CR_{Inter}$ | $CR_{Intra}$ |
|---|---|---|---|---|
| iBOT | 38.7 | 69.9 | 1.80 | 0.17 |
| +$DC_{Rec}$ | 39.3 | 69.3 | 1.32 | 0.15 |
| +$DC_{Ran}$ | 39.9 | 70.9 | 1.25 | 0.10 |
| +$DC_{Blc}$ | 41.1 | 72.9 | 1.28 | 0.09 |
| +$DC_{Cutout}$ | 39.2 | 67.7 | 1.26 | 0.10 |
| +$DC_{RCC}$ | 42.0 | 74.1 | 1.16 | 0.04 |

introduced in Section 3. Our method, when combined with all of these masking techniques, demonstrates substantial improvements in de-coupling, as evidenced by the enhanced detection and OKNN accuracy. This provides strong validation for the effectiveness of our loss design. Notably, the original Cutout method, which tends to overcut, leads to a degradation in recognition and classification performance. Among all the augmentation techniques tested, $DC_{RCC}$ consistently achieves the best performance across all metrics, further affirming the overall effectiveness of our strategy.

## 6.2 Loss Comparison

In Table 5, we conduct a comparative analysis of our de-coupling loss with the objectives employed in InsLoc (Yang et al., 2021) and CP$^2$ (Wang et al., 2022a). These methods also utilize the copy-paste strategy with a rectangular mask to define the foreground region. The primary distinction among these approaches lies in their strategies for aligning positive pairs during the pre-training stage. Specif-

Table 5: **Comparison with other copy-paste objectives**. iBOT is used as baseline.

| Method | COCO Det. | OKNN | $CR_{Inter}$ | $CR_{Intra}$ |
|---|---|---|---|---|
| iBOT | 38.7 | 69.9 | 1.80 | 0.17 |
| +$DC_{Avg}$ | 38.0 | 69.0 | 1.51 | 0.36 |
| +$DC_{p-a}$ | 37.2 | 66.7 | 1.48 | 0.45 |
| +$DC_{p-p}$ | 39.3 | 69.3 | 1.32 | 0.15 |

ically, InsLoc employs average pooling on the foreground region, denoted as $DC_{avg}$. CP$^2$ matches each point feature from the query view to all the foreground point features from the key view, termed as $DC_{p-a}$ (point-to-all). In contrast, our approach simply matches features with the same position, denoted as $DC_{p-p}$ (point-to-point). It's important to note that we use rectangular masks for all types of loss for fairness in this comparison. Both $DC_{avg}$ and $DC_{p-a}$ result in inferior intra-object coupling and a decrease in performance on downstream tasks. We attribute this to the observation that dense-level SSL benefits from fine-grained and accurate positive pair matching, rather than employing a global or coarse-grained matching strategy. Our point-to-point approach aligns better with the nature of dense SSL, leading to improved coupling and performance.

## 6.3 Augmentation: Applying RCC to Mask-based SSL

To demonstrate the effectiveness of RCC as an augmentation for both CNNs and ViTs, we select SoCo which utilizes proposal-level Cutout, and iBOT which utilizes Blockwise mask, and replace their original mask strategy with RCC. Each model is pre-trained on COCO for 800 epochs without the de-coupling branch. As shown in Tab. 6, all models using RCC outperform those

Table 6: **Comparison of mask types for dense SSL.** We replace existing mask strategies of SoCo and iBOT with RCC, denoting as '-RCC'.

| Method | COCO Det. | COCO ISeg. | OKNN | $CR_{Inter}$ |
|---|---|---|---|---|
| SoCo | 38.5 | 33.4 | 73.5 | 1.15 |
| SoCo-RCC | 38.9 | 33.9 | 74.4 | 1.10 |
| iBOT | 42.3 | 37.0 | 75.2 | 1.43 |
| iBOT-RCC | 43.1 | 37.6 | 76.8 | 1.35 |

employing the original strategy, with a lower coupling ratio, verifying the strength of RCC.

## 7 Conclusion

In this paper, we have dedicately designed experiments that have unveiled the dual impact of augmentation, both as a tool for enhancing feature representations and as a cause of feature coupling. To tackle this challenge, we introduced RCC, a novel augmentation technique, alongside a dedicated de-coupling branch for dense SSL pre-training. Our extensive experiments and comprehensive ablation studies have confirmed the superiority of RCC and our de-coupling strategy within dense SSL frameworks, and we have shown that our de-coupling strategy seamlessly integrates into various frameworks without imposing excessive computational overhead.

ACKNOWLEDGMENTS

This work was supported in part by the National Natural Science Foundation of China under Grant No. 62376209 and the Swiss National Science Foundation via the Sinergia grant CRSII5-180359.

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

# A  ALGORITHM

## A.1  ORIGINAL CUTOUT FOR DENSE SSL

---
**Algorithm 1** Proposal-level Cutout (DeVries & Taylor, 2017)
---
**Input:** Proposals $\{\boldsymbol{b}_i = (x_{i1}, y_{i1}, x_{i2}, y_{i2})\}_{i=0}^{K}$, cutout ratio $r_c$.
**Output:** Masked positions $\mathcal{M}$
  $\mathcal{M} \leftarrow \{\}$
  **for** $i < K$ **do**
    $t \leftarrow \text{Rand}(x_{i1} - d, x_{i2}); l \leftarrow \text{Rand}(y_{i1} - d, y_{i2})$
    $\mathcal{M} \leftarrow \mathcal{M} \cup \{(x, y) : x \in [t, t + a), y \in [l, l + b)\}$        $\triangleright$ Performing cutout
  **end for**
---

## A.2  REGION COLLOABORATIVE CUTOUT

---
**Algorithm 2** Region Collaborative Cutout
---
**Input:** Proposals $\{\boldsymbol{b}_i = (x_{i1}, y_{i1}, x_{i2}, y_{i2})\}_{i=0}^{K}$, cutout ratio $r_c$.
**Output:** Masked positions $\mathcal{M}$
  $\mathcal{M} \leftarrow \{\}$
  $\mathcal{B} \leftarrow \text{Sort}(\{\boldsymbol{b}_i\})$        $\triangleright$ Region set of proposals sorted from large to small
  **for** $i < K$ **do**
    $r_{ci} \leftarrow \frac{|\mathcal{B}(i)| - |\mathcal{M} \cap \mathcal{B}(i)|}{|\mathcal{B}(i)|}$        $\triangleright$ Actual ratio of $\mathcal{B}(i)$ being covered
    **if** $r_{ci} > r_c$ **then**
      $\mathcal{M} \leftarrow \mathcal{M} \setminus \{(x, y) : x \in [x_{i1}, x_{i2}), y \in [y_{i1}, y_{i2})\}$    $\triangleright$ Recover overcut proposal
      $r_{ci} \leftarrow 0$
    **end if**
    $a \leftarrow (x_{i2} - x_{i1}) \cdot (r_c - r_{ci}); b \leftarrow (y_{i2} - y_{i1}) \cdot (r_c - r_{ci})$
    $t \leftarrow \text{Rand}(x_{i1} - a, x_{i2}); l \leftarrow \text{Rand}(y_{i1} - b, y_{i2})$
    $\mathcal{M} \leftarrow \mathcal{M} \cup \{(x, y) : x \in [t, t + a), y \in [l, l + b)\}$        $\triangleright$ Performing cutout
  **end for**
---

# B  DETAILS FOR MEASURING COUPLING

**Inter-object coupling.** As illustrated in Fig 6, we calculate the $CR$ for the backbone pre-trained with various methods. For CNN-based models, we reduce the scale of object A with different reduction ratios. For ViT-based models, we apply random masking on patches of object A with different mask ratios. To extract the region-level features, we apply RoIAlign on the final feature map output by the pre-trained backbone with given bounding boxes. All the features used for calculating $CR$ values are extracted from images that are sampled from COCO *train2017* with single object.

**Intra-object coupling.** The $CR$ histograms of iBOT (Zhou et al., 2022) and DenseCL (Wang et al., 2021) are reported in Fig 3(b), and we also report the results of SoCo(Wei et al., 2021) and MaskAlign(Xue et al., 2023) in Fig. 7. The object part $\mathcal{M}_{op}$ is the intersection of a randomly selected quarter region on the bounding box and the segmentation mask of the sampled object. For feature extraction, we perform average pooling with mask $\mathcal{M}_{op}$ on the final feature map. The $CR$ values are calculated with images sampled from COCO *train2017* with ground-truth annotations.

# C  THE CAUSES OF COUPLING

A challenging pretext benefits image-level pre-training for learning more generalizable semantics for curated datasets and images containing one object. However, the same rule does not apply to dense SSL, and we demonstrate that challenging pretext tasks can lead to coupling in dense SSL.

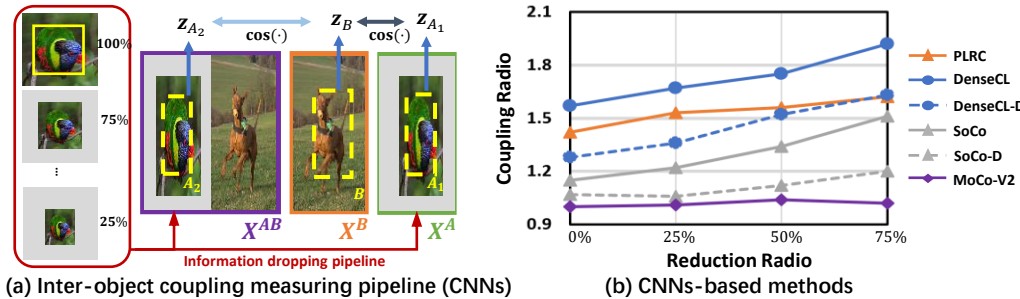

Figure 6: **Measuring inter-object coupling on CNNs.** (a) For CNN-based models, we reduce the scale of object A with different ratios to calculate the $CR$. (b) We report the $CR$ of CNN-based models. '-D' denotes method combined with our decoupling strategy.

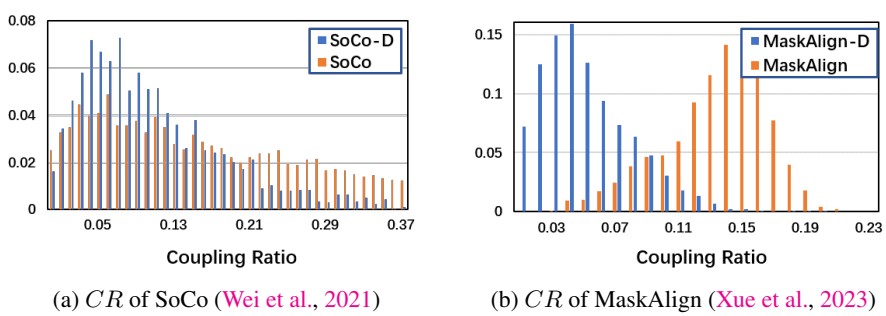

(a) $CR$ of SoCo (Wei et al., 2021)      (b) $CR$ of MaskAlign (Xue et al., 2023)

Figure 7: **Intra-object coupling of SoCo and MaskAlign.** We draw the histogram in terms of the coupling ratio. '-D' denotes those methods combined with our de-coupling strategy.

## C.1 STRONGER AUGMENTATION.

In image-level SSL, data augmentation helps to drop redundancy and keep the most discriminative information for an image. Nevertheless, it is necessary to keep essential semantic information for aligning $z_{\mathcal{P}q}^q$ with $z_{\mathcal{P}k}^k$. As an illustrative example, we employ masking with different sizes to verify that the coupling is caused by strong augmentation. Specifically, we conduct experiments with two mask-involving methods: SoCo which utilizes Cutout, and iBOT with Blockwise Mask. We pre-train each model on COCO for 200 epochs with varying mask ratios and use the inter-object $CR$ as an indicator to assess the extent of coupling. For iBOT, we calculate $CR$ on masked patches. As shown in Fig. 8, a strong correlation exists between the masking ratio and extent coupling, which validates our assumption.

## C.2 ALIGNING STRATEGY.

Apart from masking complexity, we investigate the influence of alignment accuracy on coupling. We pair features with varying alignment accuracies to demonstrate how misalignment affects coupling in DenseCL and iBOT.

**DenseCL.** For a pair of views $X^q, X^k$, and corresponding point-level features $\{z_i^q\}_{i=1}^N, \{z_i^k\}_{i=1}^N$, DenseCL aligns each $z_i^q$ with the nearest one in the key view:

$$\mathcal{L}_{DenseCL}(z_i^q) = \mathcal{L}(z_i^q, z_j^k), \; j = \arg\max_l \cos(z_i^q, z_l^k). \tag{8}$$

To establish an imprecise alignment, we align each $z_i^q$ to its $n$-th nearest key, denoted as S-T/@$n$ (S-T/@1 represents the original strategy). We also provide the result with S-T/Rand that randomly aligns each query to arbitrary key.

**iBOT** aligns the feature $z_i^q$ of each masked patch with the teacher output $z_i^k$ at the same position as the reconstruction loss, which we denote as T@1. For a less precise aligning strategy denoted as T@$n$, we align the $z_i^q$ with $z_l^k$, with $z_l^k$ being the $n$-nearest neighbor of $z_i^k$.

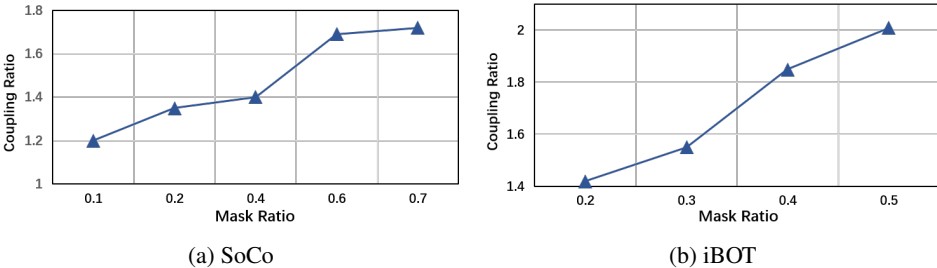

|  |  |
|:---:|:---:|
| (a) SoCo | (b) iBOT |

Figure 8: **Coupling ratio with different Mask Ratio.** Both SoCo and iBOT are pre-trained on COCO for 200 epochs with varying mask ratios. 'Coupling Ratio' denotes the inter-object $CR$.

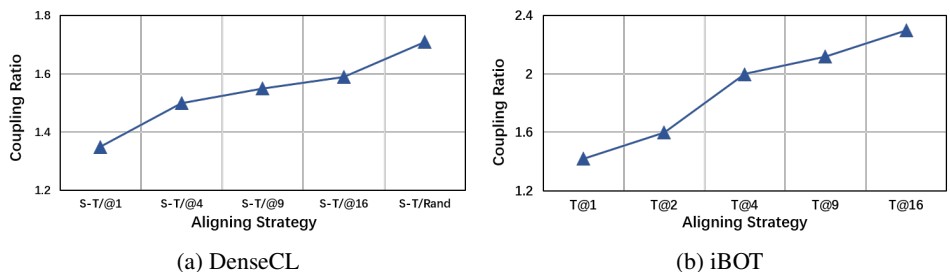

|  |  |
|:---:|:---:|
| (a) DenseCL | (b) iBOT |

Figure 9: **Coupling ratio with different aligning strategy.** 'Coupling Ratio' denotes the inter-object $CR$. The horizontal axis of each method represents the increasingly imprecise aligning strategies from left to right.

In Fig. 9, we present the inter-object $CR$ of each model pre-trained on COCO for 200 epochs, illustrating that less accurate alignment of positive pairs tends to lead to more significant coupling.

## D    MORE ANALYSIS OF MASKS

As one of the core designs of our de-coupling strategy, the proposed RCC is designed specifically for dense SSL. We thus further explore the effects of RCC compared to other masking strategies.

### D.1    RCC VS. CUTOUT

RandomResizeCrop plays a crucial role in image-level SSL and enhances the network's ability to capture robust semantic representations, however, there has been a lack of region-specific masking strategies for dense SSL. This naturally leads to the idea of Cutout(DeVries & Taylor, 2017), which randomly cuts out square regions of the image for each region. However, there are typically overlapping areas of cutout between adjacent regions, which results in over-cutout, i.e., regions that are largely covered by the cutout from other regions. To verify the effectiveness of our method in solving this issue, we provide an additional experiment. Specifically, we generate $N$ RCC and Cutout masks with a cutout ratio of 0.4 and calculate the average proportion of the actual cutout area in each proposal region. As shown in Fig. 11, when the number of proposals in each image increases, Cutout masks suffer from severe over-cutout issues, and the actual cutout ratio is much greater than 0.4. Our RCC stabilizes at the preset cutout ratio.

### D.2    COMPARSION OF DIFFERENT MASKING STRATEGIES

To illustrate the distinctions between masking strategies, we employ four different methods for generating masks, as depicted in Fig. 12. We partition an image into $2 \times 2$ regions and generate one bounding box for each region with a predefined cut ratio. It's important to note that the center of each bounding box is randomly selected within its respective region, potentially causing overlap between

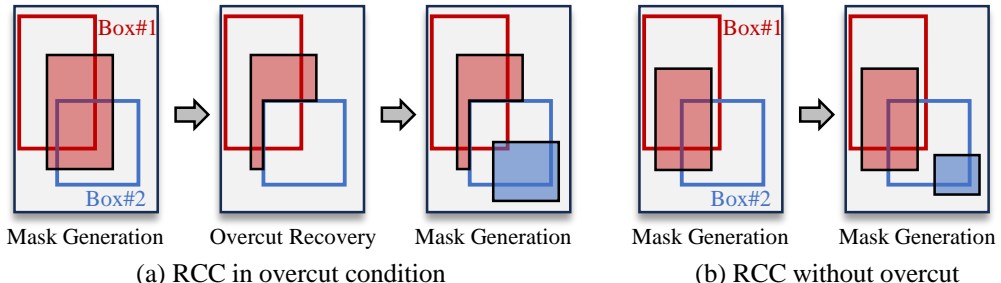

(a) RCC in overcut condition      (b) RCC without overcut

Figure 10: **Illustration of RCC algorithm.** RCC iteratively recovers the overcut region caused by the larger bounding boxes. We display the example with 2 boxes.

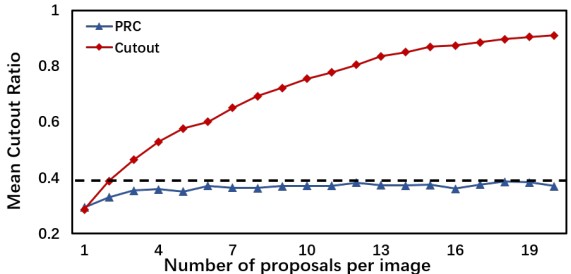

Figure 11: **Actual cutout ratio of RCC and Cutout in different numbers of regions.** We set the cutout ratio for each region to 0.4 for both methods. The X-axis is the number of proposed regions and Y-axis is the average cutout ratio. All the cutouts are applied to each region, which is generated randomly with scale [0.2, 1.0] and aspect ratio [0.5, 2], and mask size is set to $224 \times 224$. The experiment is repeated 500 times.

masks from different regions. In comparison to the other three methods, our approach generates masks with fewer overlaps and more extensive coverage across the entire image.

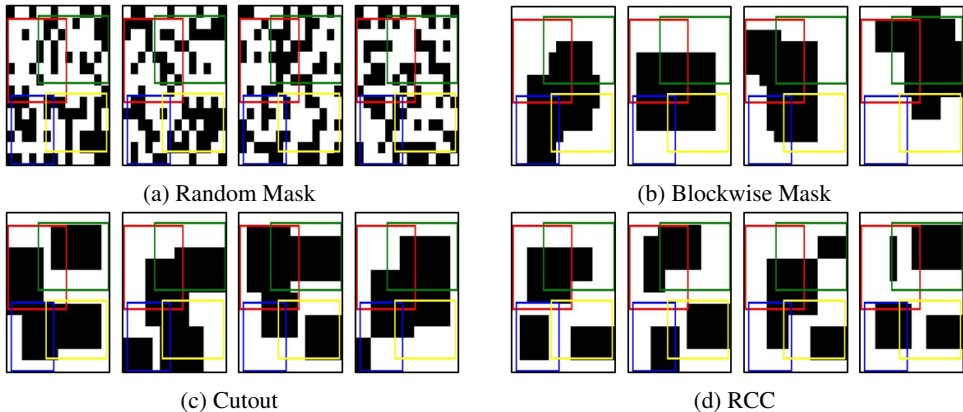

(a) Random Mask         (b) Blockwise Mask

(c) Cutout          (d) RCC

Figure 12: **Examples of different masking strategies.** We generate masks utilizing different strategies with a mask ratio of 0.4. For Cutout and RCC, we generate regions on $2 \times 2$ grids.

# E  IMPLEMENTATION DETAILS

## E.1  DENSE-LEVEL OBJECTIVE WITH DE-COUPLING

We will explain how we incorporate the de-coupling strategy with different methods.

**Point-level/Patch-level methods.** Point-level methods directly perform SSL on each point-level feature from the output feature map. For de-couping, we first generate a de-coupled view $X^D$ with

the background mask $\mathcal{M}_{RCC}$ and the key view $\boldsymbol{X}$. We align the point-level features locating on the foreground region with the output of teacher model with corresponding coordinate. The implementations of de-coupling on DenseCL(Wang et al., 2021), iBOT(Zhou et al., 2022) and MaskAlign(Xue et al., 2023) follow the above setting. For brevity, we define the mask as the index set of patches it covers in the case of point-level methods. As the details:

a) **DenseCL** implements infoNCE loss on pair-wise point-level features extracted from view $X^q$ and $X^k$, which leads to objective:

$$\mathcal{L}_{DenseCL} = (1 - \lambda)\mathcal{L}_{NCE}(\boldsymbol{q}, \boldsymbol{k}, \{\boldsymbol{k}_-\}; \tau) + \frac{\lambda}{|\{\boldsymbol{q}^i\}|} \sum_i \mathcal{L}_{NCE}(\boldsymbol{q}^i, \boldsymbol{k}^i, \{\boldsymbol{k}_-^i\}; \tau), \qquad (9)$$

where $\mathcal{L}_{NCE}$ is the infoNCE loss with temperature $\tau$. Denote point-level features from view $\boldsymbol{X}^D$ as $\boldsymbol{k}_D^j$, the de-coupling version of DenseCL is then defined as:

$$\mathcal{L}_{DenseCL-D} = (1 - \lambda_{DC})\mathcal{L}_{DenseCL} + \frac{\lambda_{DC}}{|\mathcal{M}_{RCC}^C|} \sum_{j \in \mathcal{M}_{RCC}^C} \mathcal{L}_{NCE}(\boldsymbol{k}_D^j, \boldsymbol{k}^j, \{\boldsymbol{k}_-^j\}; \tau). \qquad (10)$$

b) **iBOT** adopts the same image-level distillation loss as DINO (Caron et al., 2021), but introduces a novel pixel-level distillation loss on masked patches, as a reconstruction loss. In this paper, we apply iBOT without multi-crop. This leads to the objective:

$$\mathcal{L}_{iBOT} = -f_t^{[CLS]}(\boldsymbol{X}^k)^T \log f_s^{[CLS]}(\boldsymbol{X}^q) - \frac{1}{|\mathcal{M}|} \sum_{i \in \mathcal{M}} f_t(\boldsymbol{X}^k, \mathcal{P}_i)^T \log f_s(\boldsymbol{X}^M, \mathcal{P}_i), \qquad (11)$$

where $\mathcal{M}$ represents the mask for reconstruction. The masked view, denoted as $\boldsymbol{X}^M$, is created using $\boldsymbol{X}^k$, and $\mathcal{P}_i$ indicates the position of each patch. We construct the de-coupled view $\boldsymbol{X}^D$ with $\boldsymbol{X}^k$, which leads to iBOT-D as :

$$\mathcal{L}_{iBOT-D} = (1 - \lambda_{DC})\mathcal{L}_{iBOT} - \frac{\lambda_{DC}}{|\mathcal{M}_{RCC}^C|} \sum_{j \in \mathcal{M}_{RCC}^C} f_t(\boldsymbol{X}^k, \mathcal{P}_j)^T \log f_s(\boldsymbol{X}^D, \mathcal{P}_j). \qquad (12)$$

c) **MaskAlign** applies a pre-trained encoder as a frozen teacher and directly aligns the visible features with a smooth L1 loss to the corresponding features output by the teacher model. This leads to the objective:

$$\mathcal{L}_{L1}(\boldsymbol{z}_1, \boldsymbol{z}_2) = \begin{cases} \frac{1}{2}\|\boldsymbol{z}_1 - \boldsymbol{z}_2\|_2, & \|\boldsymbol{z}_1 - \boldsymbol{z}_2\|_1 \leq 1 \\ \|\boldsymbol{z}_1 - \boldsymbol{z}_2\|_1 - \frac{1}{2}, & \text{otherwise} \end{cases}, \qquad (13)$$

$$\mathcal{L}_{MaskAlign} = \frac{1}{|\mathcal{M}^C|} \sum_{i \notin \mathcal{M}} \mathcal{L}_{L1}\left[f_s(\boldsymbol{X}^M, \mathcal{P}_i), f_t(\boldsymbol{X}, \mathcal{P}_i)\right]. \qquad (14)$$

We then generate de-coupled view $\boldsymbol{X}^D$ from $\boldsymbol{X}$ and combine de-coupling strategy as

$$\mathcal{L}_{MaskAlign-D} = (1 - \lambda_{DC})\mathcal{L}_{MaskAlign} + \frac{\lambda_{DC}}{|\mathcal{M}_{RCC}^C|} \sum_{j \in \mathcal{M}_{RCC}^C} \mathcal{L}_{L1}\left[f_s(\boldsymbol{X}^D, \mathcal{P}_j), f_t(\boldsymbol{X}, \mathcal{P}_j)\right].$$
$$(15)$$

**Region-level methods.** Region-level methods define regions as the collections of point-level features, serving as object priors. To incorporate with such setting, we perform our de-coupling loss on the intersection of the selected region and foreground $\mathcal{M}_{RCC}^C$. The implementations of SoCo(Wei et al., 2021) and Leopart(Ziegler & Asano, 2022) follow the above setting. Specifically:

a) **SoCo** apply two types of framework for extracting region-level features: an R50-C4 structure that leverages $conv5$ block with RoiAlign on feature map output by $conv4$; an R50-FPN structure that extracts multi-scale region-level features with an extra FPN. We adopt R50-C4 for its simplicity.

Note that SoCo applies selective search to generate $K$ proposals for each view as object prior, we thus use the same proposals to obtain background mask $\mathcal{M}_{RCC}$. For two augmented view $X^q, X^k$ and corresponding proposal region $\{\mathcal{P}_i^q\}_{i=1}^K, \{\mathcal{P}_i^k\}_{i=1}^K$, the objective of SoCo-C4 is defined as:

$$\mathcal{L}_{SoCo} = \frac{1}{K} \sum_i \cos\left(f_s(\boldsymbol{X}^q, \mathcal{P}_i^q), f_t(\boldsymbol{X}^k, \mathcal{P}_i^k)\right). \tag{16}$$

We extract region-level de-coupled features on the intersection of $\mathcal{M}_{RCC}^C$ and each proposal, $i.e.$:

$$\mathcal{L}_{SoCo-D} = (1 - \lambda_{DC})\mathcal{L}_{SoCo} + \frac{\lambda_{DC}}{K} \sum_j \cos\left(f_s(\boldsymbol{X}^D, \mathcal{P}_j^k \cap \mathcal{M}_{RCC}^C), f_t(\boldsymbol{X}^k, \mathcal{P}_j^k)\right). \tag{17}$$

b) **Leopart** applies a SwAV-like prototype-based loss(Caron et al., 2020) on region-level features. It generates $V$ views with $v_g$ global views and aligns region-level features masked by the global view's attention, which leads to the objective with 2D cross-entropy loss $l$ as:

$$\mathcal{L}_{Leopart} = \sum_{j=0}^{v_g} \sum_{i=0}^{V} \mathbf{1}_{i \neq j} \left[\boldsymbol{M}_{a_{ij}} \odot l\left(f_s(\boldsymbol{X}_i, \mathcal{P}_i), f_t(\boldsymbol{X}_j, \mathcal{P}_j)\right)\right], \tag{18}$$

where $\boldsymbol{M}_{a_{ij}}$ is the binary mask of global view's attention to the intersection with view $\boldsymbol{X}_j$. $\mathcal{P}_i, \mathcal{P}_j$ is the corresponding pairwise intersection between $\boldsymbol{X}_i, \boldsymbol{X}_j$. Because local views typically provide incomplete semantics, We only perform de-coupling on global views to avoid a difficult pretext that leads to coupling. With the de-coupled view $\boldsymbol{X}_j^D$ generated by the global view $\boldsymbol{X}_j$, we define the de-coupling objective as:

$$\mathcal{L}_{Leopart-D} = (1 - \lambda_{DC})\mathcal{L}_{Leopart} + \lambda_{DC} \sum_{j=0}^{v_g} (\boldsymbol{M}_{a_j} \odot \boldsymbol{M}_{F,j}) \odot l\left(f_s(\boldsymbol{X}_j^D, \mathcal{I}), f_t(\boldsymbol{X}_j, \mathcal{I})\right), \tag{19}$$

where the binary mask $\boldsymbol{M}_{F,j}, \boldsymbol{M}_{a_j}$ are the foreground defined by RCC and attention mask corresponding to $\boldsymbol{X}_j$ respectively.

## E.2 PRE-TRAINING DETAILS

**De-coupling.** For region generation and RCC mask, we divide the input view into $3 \times 3$ grids and create a single bounding box in each grid with the scale in $(0.15, 4)$ and aspect ratio in the range $(0.5, 2)$. The RCC cutout ratio is selected from the range $[0.3, 0.5]$. The de-coupled views are generated with the augmented key views, which are processed with the standard augmentation pipeline such as crop, jittering and gaussian filtering. We set the de-coupling weight $\lambda_{DC}$ to 0.3.

**Optimization hyper-parameters.** For DenseCL, we follow the same setting to adopt an SGD optimizer with the base learning rate $lr_{base}$ of 0.3. For SoCo, we adopt LARS (You et al., 2017) with $lr_{base}$ as 2.0 and batch size of 1024. For Leopart, we keep all the same. For iBOT, we utilize AdamW (Loshchilov & Hutter, 2017) and set the $lr_{base}$ to $1 \times 10^{-3}$ and the batch size of 512. For MaskAlign, we use AdawW with $lr_{base} = 3 \times 10^{-4}$ and the batch size of 1024.

**Pre-training schedule.** For models with ResNet-50, to compare with the existing baseline, we follow the common schedule of 800-epoch pre-training adopt in (Wang et al., 2021; Bai et al., 2022). For models with ViT-S/16, since we mainly focus on the comparisons between same models with and without de-coupling, we adopt the 800-epoch pre-training schedule for both iBOT and MaskAlign.

## E.3 EVALUATION PROTOCOLS.

Here, we will explain the implementation details of the evaluation protocols in our paper.

Table 7: **Semantic segmentation results of model pre-trained with Leopart.** '-D' denotes combining with de-coupling strategy, 'LC' denotes finetuning a linear classifier, 'FCN' denotes finetuning an FCN model, and 'K=500' denotes over-clustering with K set as 500.

| Method | VOC12 | | | COCO-Stuff | |
|---|---|---|---|---|---|
| | LC | FCN | K=500 | LC | K=500 |
| Leopart | 65.0 | 67.2 | 45.2 | 52.5 | 44.4 |
| Leopart-D | 66.5 | 68.9 | 48.2 | 53.2 | 45.3 |

### E.3.1 FINETUNING FOR DOWNSTREAM TASKS

**CNN-based evaluation** We compare models using a ResNet-50 backbone on three dense-level benchmarks: Pascal VOC detection, COCO detection, and instance segmentation. For VOC detection, we fine-tune a Faster R-CNN detector (C4-backbone) on the combined set of *trainval2007* and *trainval2012*. As the variation on Pascal VOC is large, following (He et al., 2020), we run 5 trials and compute means. We then report AP, AP50, and AP75 on the *test2007* set. For COCO detection and instance segmentation, we fine-tune a Mask R-CNN detector (C4-backbone) on COCO *train2017* with $1\times$ schedule. The evaluation is performed on the COCO *val2017* split.

**ViT-based evaluation** We compare models using a ViT-S/16 backbone on three benchmarks: COCO detection and instance segmentation, and ADE20K semantic segmentation following (Zhou et al., 2022). For COCO detection and segmentation, we fine-tune a Cascade Mask R-CNN framework with a $1\times$ schedule on the *train2017* split. For ADE20K segmentation, we adopt the task layer in UPerNet and fine-tune the entire network for 160k iterations on the *training* split.

### E.3.2 DETAILS OF OKNN AND OKNN-D

**OKNN.** We obtain object-level features from the output feature map by using ground-truth bounding boxes. We achieve this using RoI Align with a size of $7 \times 7$ and average pooling. Specifically, we resize each input image to $224 \times 224$. For ResNet50, we upsample the output of $conv4$ to $28 \times 28$ and obtain a $14 \times 14$ feature map output by $conv5$. Meanwhile, we directly utilize the $14 \times 14$ output of ViT-S/16. For the training set, we extract $N$ object-level features per image to store the features and labels in a memory bank because choosing all the objects in the dataset will lead to imbalanced numbers across the categories in the training set, which will lead to the results being biased. Similarly to image-level KNN, we predict the label for each object in the evaluation set by finding the $k$-nearest object-level features in the training set. For evaluation on COCO, we set $N = 3$ and $k = 20$. The above process does not need further training and is much faster. More importantly, compared to the evaluation with a fine-tuning stage, O-KNN directly reflects how well the features from the same class are preserved instead of favoring the mask or detection prediction.

**OKNN-D.** OKNN-D Follows the same pipeline for feature extraction and label prediction as OKNN. The difference is that for every input image, we directly replace the regions uncovered by the selected bounding boxes with arbitrary background images. When a pre-trained model encounters severe coupling issues, it tends to intertwine each selected object with irrelevant background bias, resulting in a more significant degradation of recognition.

## F ADDITIONAL RESULTS

**Evaluation of Leopart.** The evaluations of Leopart(Ziegler & Asano, 2022) follow its original protocols. Specifically, we utilize three techniques for evaluation: a) linear classifier (LC): we fine-tune a $1 \times 1$ convolutional layer on top of the spatial output of the frozen backbone following (Van Gansbeke et al., 2021); b) over-clustering, we evaluate the frozen feature using spatially dense clustering and Hungarian matching (Kuhn, 1955); c) FCN, we fine-tune a FCN model (Long et al., 2015) following same setting as (Wang et al., 2021). We report results on Pascal VOC and COCO-Stuff in Tab. 7. Our model advances original Leopart significantly in transfer learning on VOC. On COCO-Stuff, Leopart-D also achieves an improvement of 0.7 on LC and 0.9 on over-clustering. Note that

Table 8: **Training time of different models.** We detail the actual training time of different models. Each model is pre-trained on COCO for 800 epochs with an 8-GPU 3090 machine. B denotes the batch size, '-D' denotes models with de-coupling. For de-coupling, we sample 25% key views to construct the de-coupled views which are only forwarded to the student encoder.

| Method | Total Crops Number | Backbone | Training Time |
|---|---|---|---|
| SoCo | $B\times2\times224^2$ | ResNet-50 | 27h |
| SoCo-D | $B\times2\times224^2+B\times0.25\times224^2$ | ResNet-50 | 30h |
| DenseCL | $B\times2\times224^2$ | ResNet-50 | 29h |
| DenseCL-D | $B\times2\times224^2+B\times0.25\times224^2$ | ResNet-50 | 32h |
| iBOT | $B\times2\times224^2$ | ViT-S/16 | 24h |
| iBOT-D | $B\times2\times224^2+B\times0.25\times224^2$ | ViT-S/16 | 27h |
| MaskAlign | $B\times1\times224^2$ | ViT-S/16 | 10h |
| MaskAlign-D | $B\times1\times224^2+B\times0.25\times224^2$ | ViT-S/16 | 13h |

Table 9: **Different pre-training schedules.** We report COCO detection and instance segmentation performance. '-MC' denotes iBOT with a multi-crop stratigy.

| Method | Epoch | COCO Det. | | | COCO Iseg. | | |
|---|---|---|---|---|---|---|---|
| | | AP | $AP_{50}$ | $AP_{75}$ | AP | $AP_{50}$ | $AP_{75}$ |
| iBOT | 800 | 42.3 | 61.2 | 45.6 | 37 | 58.3 | 39.4 |
| iBOT | 900 | 42.7 | 61.9 | 45.8 | 37.2 | 58.6 | 39.6 |
| iBOT-MC | 800 | 43.7 | 62.8 | 47.0 | 38.1 | 59.6 | 40.7 |
| iBOT-D | 800 | 45.1 | 64.3 | 48.7 | 39.1 | 61.2 | 41.7 |

Leopart is only pre-trained for 50 epochs, following the original setting. It demonstrates that our de-coupling strategy can also work with a short pre-training schedule.

**Training time and learning efficiency.** In Tab. 8, we provide the training times for each method when pre-trained on COCO for 800 epochs. To maintain a balance between the de-coupling effect and additional computational costs, we employ a strategy where we sample 25% of key views to create the de-coupled views without introducing extra augmentations. These de-coupled views are exclusively forwarded to the student encoder. Across most dense SSL methods, the inclusion of a de-coupling branch results in an approximately 10% increase in training time.

**Learning efficiency.** To investigate whether the observed benefits of our de-coupling strategy are solely due to an increase in the number of views, we conducted a comparison using different pre-training schedules for iBOT. a) In the first schedule, we used 25% of query views to create de-coupled views and pre-trained iBOT for 900 epochs on COCO, maintaining the same total number of views. b) In the second schedule, we employed a multi-crop strategy for iBOT's pre-training, which included two global views sized at $224^2$ and four local views sized at $96^2$, denoted as iBOT-MC. As shown in Tab. 9, our iBOT-D outperformed all other models by a significant margin, demonstrating that the advantages of our approach extend beyond the increase in the number of views.

# G ADDITIONAL ABLATIONS

## G.1 ABLATION ON CNNS

**Mask types.** In Section 6, we conducted ablation experiments to assess the impact of different components of the de-coupling strategy using iBOT. Following the same experimental setup, we pre-trained DenseCL on COCO for 200 epochs and examined the effects of our RCC and de-coupling loss. The results are presented in Table 10. For the details of the masking strategy: a) $DC_{Rec}$: We generated rectangular masks within the scale range of $[0.2, 0.6]$ and aspect ratio range of $[2/3, 3/2]$. b) $DC_{Ran}$: We randomly masked out patches as background with a mask ratio sampled from the range $[0.4, 0.6]$. c) For other methods, we sampled the mask/cutout ratio from the range $[0.3, 0.5]$. Similar to our findings with iBOT, we observed that the presence of a copy-paste branch aids in

Table 10: **Comparison of mask types.** DenseCL is used as a baseline, we only change the masking strategy for de-coupling.

| Method | VOC Det. | OKNN | $CR_{Inter}$ | $CR_{Intra}$ |
|---|---|---|---|---|
| DenseCL | 55.0 | 67.5 | 1.64 | 0.31 |
| +$DC_{Rec}$ | 55.2 | 66.7 | 1.25 | 0.24 |
| +$DC_{Ran}$ | 55.5 | 69.3 | 1.22 | 0.17 |
| +$DC_{Cutout}$ | 54.6 | 65.1 | 1.30 | 0.25 |
| +$DC_{Blc}$ | 55.3 | 69.0 | 1.16 | 0.20 |
| +$DC_{RCC}$ | 55.9 | 70.0 | 1.18 | 0.12 |

Table 11: **Comparison with copy-paste methods.** DenseCL is used as baseline.

| Method | VOC Det. | OKNN | $CR_{Inter}$ | $CR_{Intra}$ |
|---|---|---|---|---|
| DenseCL | 55.0 | 67.5 | 1.64 | 0.31 |
| +$DC_{Avg}$ | 54.4 | 65.2 | 1.33 | 0.33 |
| +$DC_{p-a}$ | 54.1 | 63.5 | 1.41 | 0.40 |
| +$DC_{p-p}$ | 55.2 | 66.7 | 1.25 | 0.24 |

de-coupling at the inter-object level. However, only masks that introduce context noise at a finer granularity can effectively mitigate intra-object coupling. Masks generated with RCC outperformed all other strategies in both downstream performance and de-coupling effects

**Loss types.** Table 11 presents a comparison of different types of de-coupling loss using a rectangular mask. The results demonstrate that aligning each decoupled dense-level feature directly to the key with the corresponding position is an effective approach without performance degradation.

## G.2 ABLATION ON HYPER-PARAMETERS

**Number of grids.** The input view is divided into $N_{grid} \times N_{grid}$ grids to generate a single bounding box in each grid. We use varying values of $N_{grid}$ and pre-train iBOT-D on COCO for 200 epochs to investigate its influence on RCC for decoupling. Results in Tab. 12 indicate a low number of grids which prevents RCC from effectively generating cutout regions in various positions and will weaken the effectiveness of de-coupling, resulting in lower performance on downstream tasks. In practice, we set $N_{grid}$ to 3 for computational efficiency.

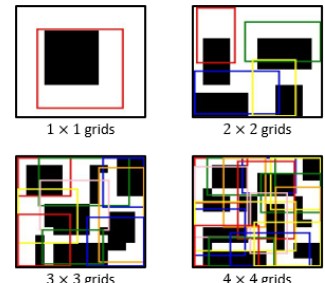

Figure 13: Background mask with different numbers of grids.

Table 12: **Ablation on the number of grids.** We report AP of COCO detection and coupling ratio with models pre-trained with different numbers of grids.

| $N_{grid}$ | 1 | 2 | 3 | 4 |
|---|---|---|---|---|
| COCO Det. | 39.4 | 41.2 | 42.0 | 42.1 |
| $CR_{Inter}$ | 1.42 | 1.10 | 1.15 | 1.14 |
| $CR_{Intra}$ | 0.16 | 0.08 | 0.04 | 0.03 |

**de-coupling weight.** We pre-train the iBOT-D with different de-coupling weights $\lambda_{DC}$ with results shown in Tab. 13. A high $\lambda_{DC}$ will disturb the original pre-training objective, leading to worse performance on downstream tasks. We thus set the $\lambda_{DC}$ to 0.3 for the balance of de-coupling and dense SSL objectives.

Table 13: **Ablation on $\lambda_{DC}$.** We report AP of COCO detection and coupling ratio with models pre-trained with different de-coupling weights $\lambda_{DC}$.

| $\lambda_{DC}$ | 0.1 | 0.2 | 0.3 | 0.4 | 0.6 |
|---|---|---|---|---|---|
| COCO Det. | 40.5 | 41.6 | 42.0 | 41.2 | 38.4 |
| $CR_{Inter}$ | 1.25 | 1.18 | 1.15 | 1.12 | 1.16 |
| $CR_{Intra}$ | 0.07 | 0.03 | 0.04 | 0.02 | 0.02 |

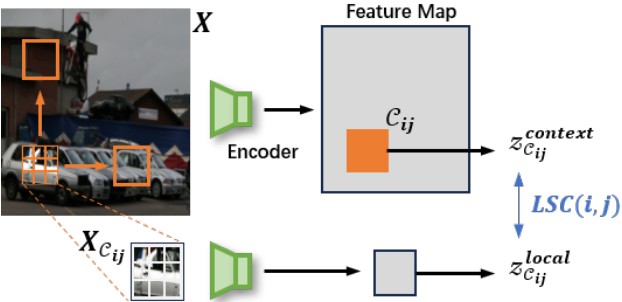

Figure 14: **Pipeline of calculating LSC.** We sample a small crop $\boldsymbol{X}_{\mathcal{C}_{ij}}$ from the origninal view $\boldsymbol{X}$ with the center lying on the patch of $i$-th row and $j$-th column. $\boldsymbol{X}_{\mathcal{C}_{ij}}$ and $\boldsymbol{X}$ are respectively forwarded to the encoder. We calculate the cosine similarity between the image-level feature of $\boldsymbol{X}_{\mathcal{C}_{ij}}$ and the corresponding region-level feature of $\boldsymbol{X}$ as the value of $LSC(i, j)$.

# H VISUALIZATION

## H.1 POINT AFFINITIES

Previous work Bai et al. (2022) provides a simple yet intuitive visualization technique to show how well the SSL model learns from an image. We adopt a similar approach to visualize the feature similarity maps generated by different pre-training methods, which intuitively demonstrates the effectiveness of our de-coupling strategy. We sample images from COCO for visualization and resize them to the size of 448×448. The feature maps are obtained by the interpolation of the output feature map from the final layer to 56×56. For each image, a specific feature was selected (marked as a red dot), and its cosine similarity to the remaining features in other locations from the feature map was calculated. The models used for visualization are pre-trained on COCO for 800 epochs. The results of CNN-based and ViT-based models are shown in Fig. 15 and 16.

## H.2 LOCAL SEMANTICS CONSISTENCY

As depicted in Fig. 14, we argue that a network effectively capturing local semantics should be able to maintain consistency with or without neighboring context. Based on this assumption, we verify the effectiveness of the de-coupling paradigm through a new type of visualization. For a given image $\boldsymbol{X}$ of size $512 \times 512$, we first patchify it with a patch size of 16, yielding $32 \times 32$ patches $\mathcal{P}_{ij}$. We define crop region $\mathcal{C}_{ij}$ whose center lies on $\mathcal{P}_{ij}$, containing $5 \times 5$ patches. The local semantics consistency map is defined as:

$$\boldsymbol{z}_{\mathcal{C}_{ij}}^{local} = E(\boldsymbol{X}_{\mathcal{C}_{ij}}), \; \boldsymbol{z}_{\mathcal{C}_{ij}}^{context} = f(\boldsymbol{X}, \mathcal{C}_{ij}), \; LSC(i, j) = \cos\left(\boldsymbol{z}_{\mathcal{C}_{ij}}^{local}, \boldsymbol{z}_{\mathcal{C}_{ij}}^{context}\right). \quad (20)$$

We directly pass a local crop, denoted as $\boldsymbol{X}_{\mathcal{C}_{ij}}$, through the encoder to obtain the local feature at $\mathcal{C}_{ij}$. Simultaneously, we send the entire image $\boldsymbol{X}$ to the backbone network $E$ and extract region-level features corresponding to the area $\mathcal{C}_{ij}$, resulting in $\boldsymbol{z}_{\mathcal{C}_{ij}}^{context}$. A higher value in the $LSC$ map shows that the representation of the local region is resilient to the absence of contextual information, rather than being overwhelmed by its neighboring context, indicating a robust local feature. The results for both CNN-based and ViT-based models are illustrated in Figure 17 and Figure 18.

## H.3 FURTHER DISCUSSION WITH DINO V2

DINO v2(Oquab et al., 2023) applies iBOT loss for dense-level pre-training. To further investigate the model's behavior of capturing local semantics when pre-trained on large-scale curated datasets, we visualize DINO v2 following the same pipeline adopted in Appendix H.1 with the officially released ViT-S/14 and ViT-B/14 distilled models. We show the failure cases in Fig. 19 where the model entangles the features of sub-discriminative objects with their surroundings. We hypothesize that even the foundation model pre-trained at a dense level without the de-coupling constraint can suffer from the coupling issue. We will leave further discussion on this issue for future work.

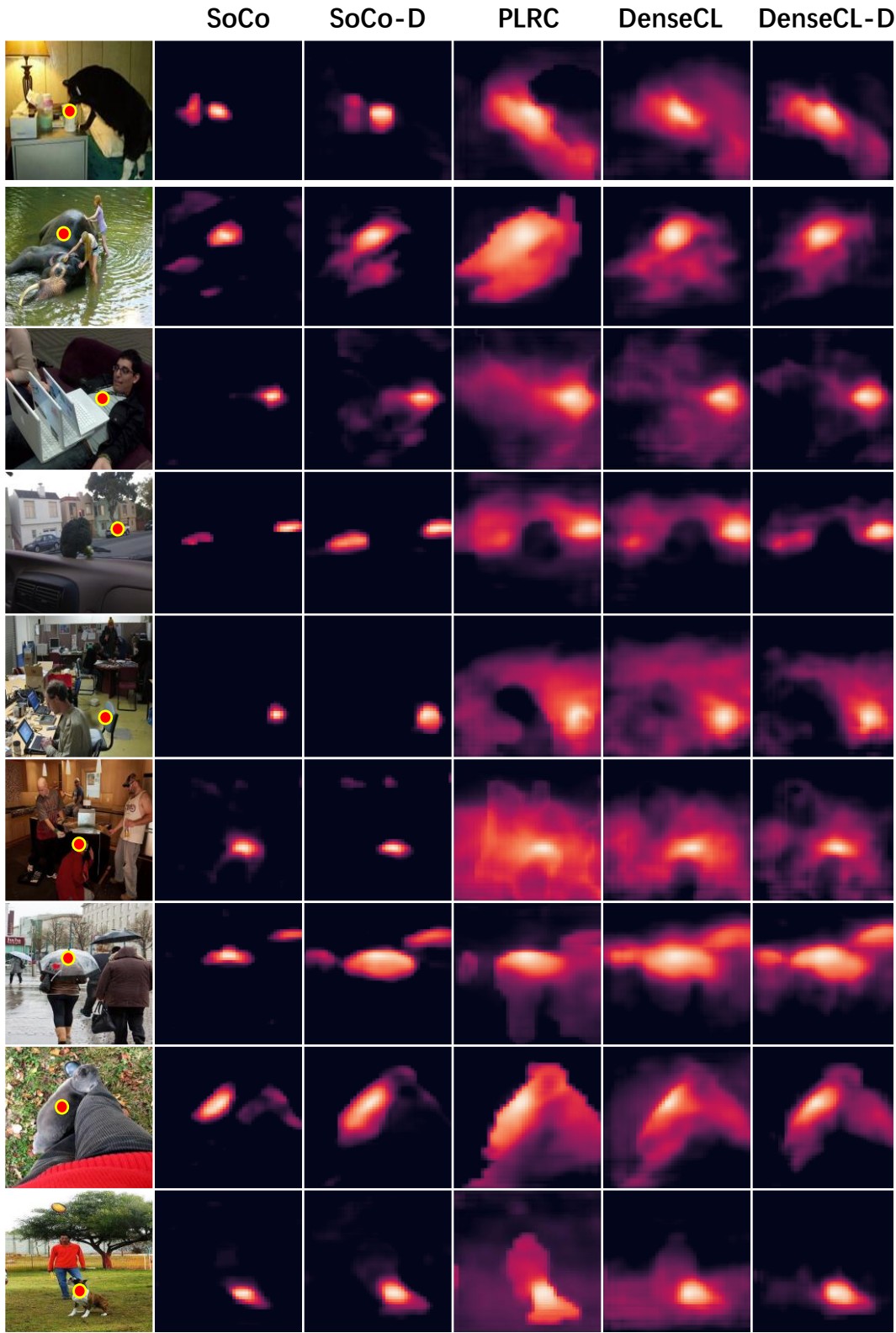

Figure 15: **Point affinity visualization on CNN-based models.** The selected point-level feature is marked as a red dot.

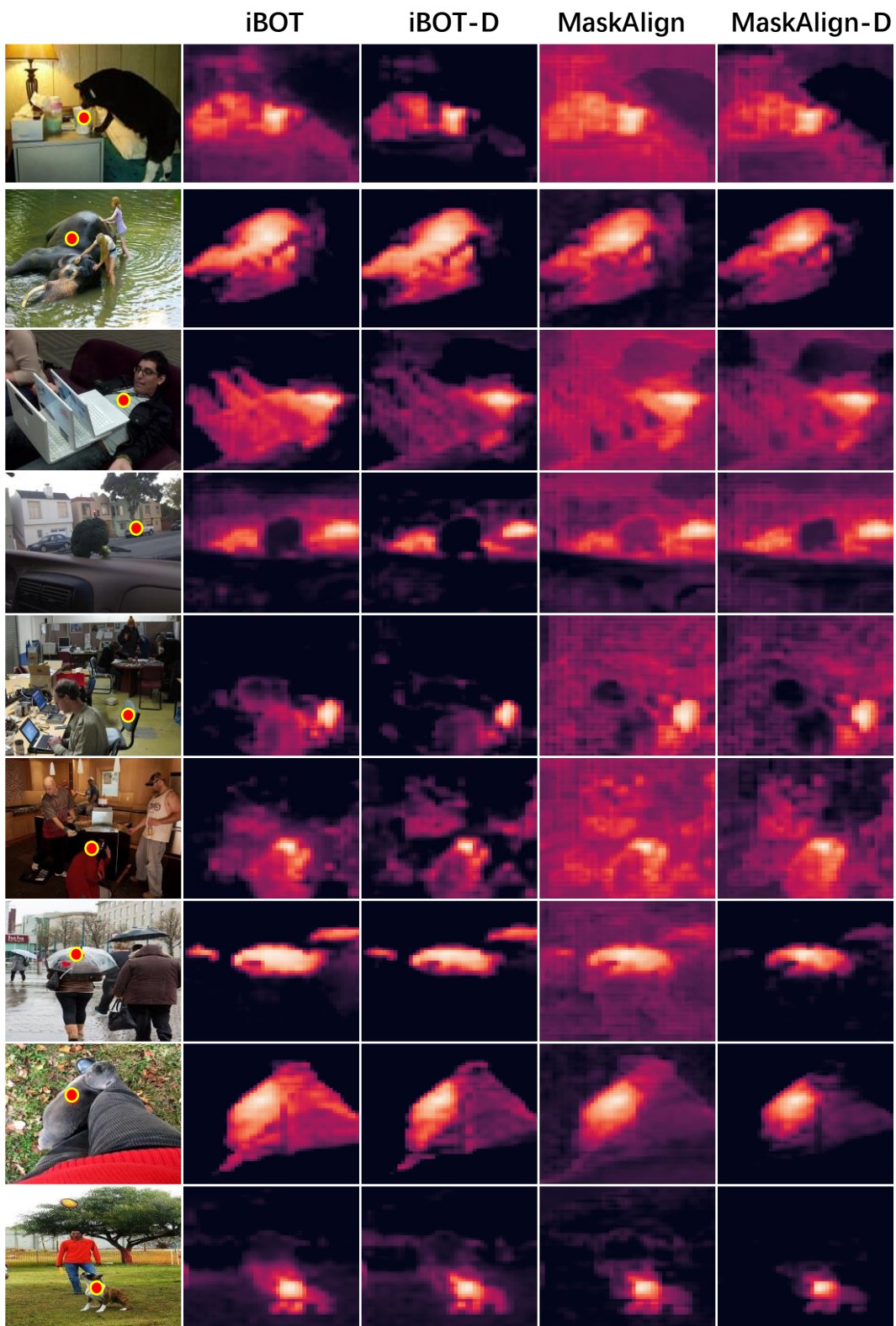

Figure 16: **Point affinity visualization on ViT-based models.** The selected point-level feature is marked as a red dot.

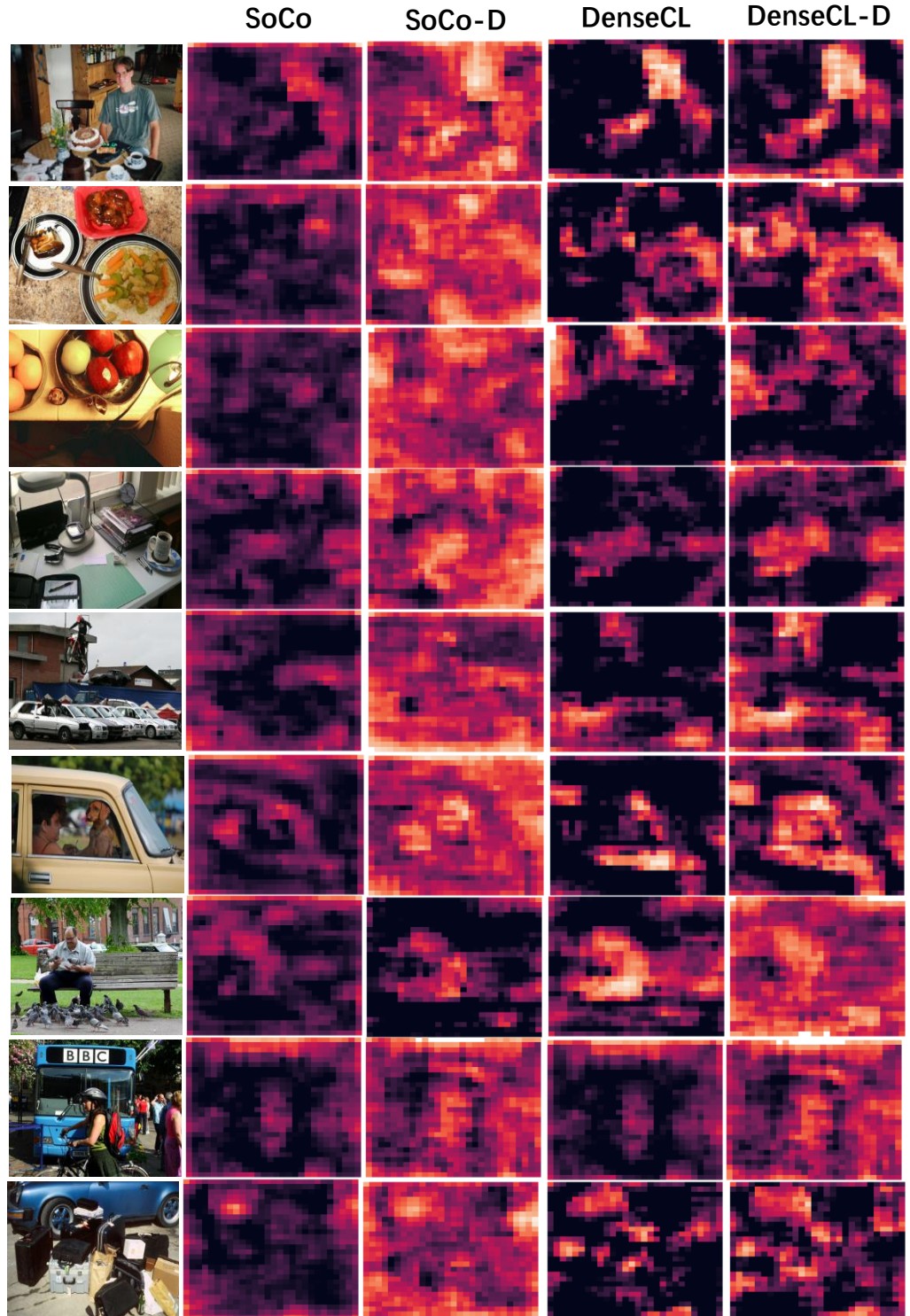

Figure 17: **Visualization of LSC map on CNN-based models.** Brighter colors denote higher LSC.

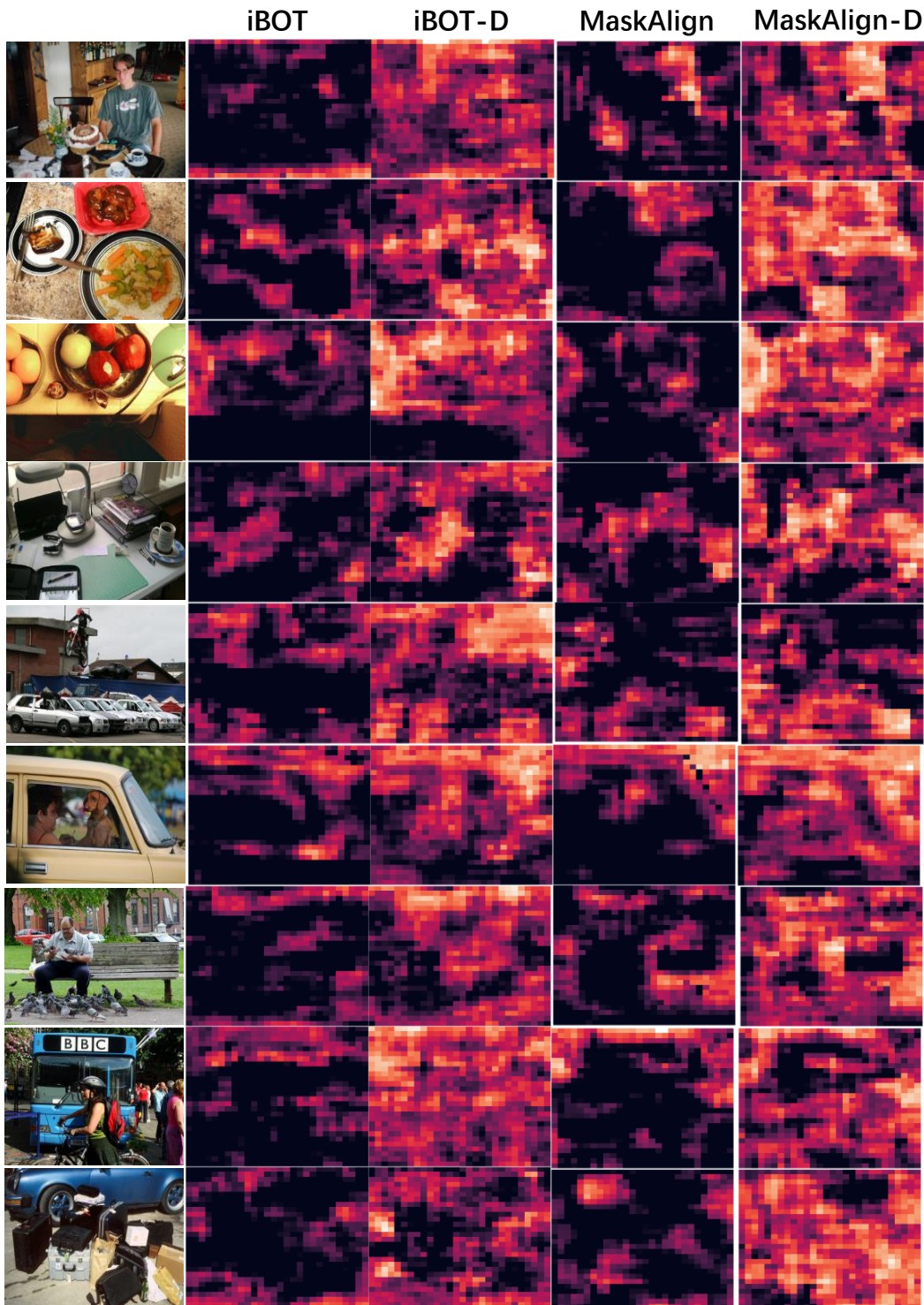

Figure 18: **Visualization of LSC map on ViT-based models.** Brighter colors denote higher LSC.

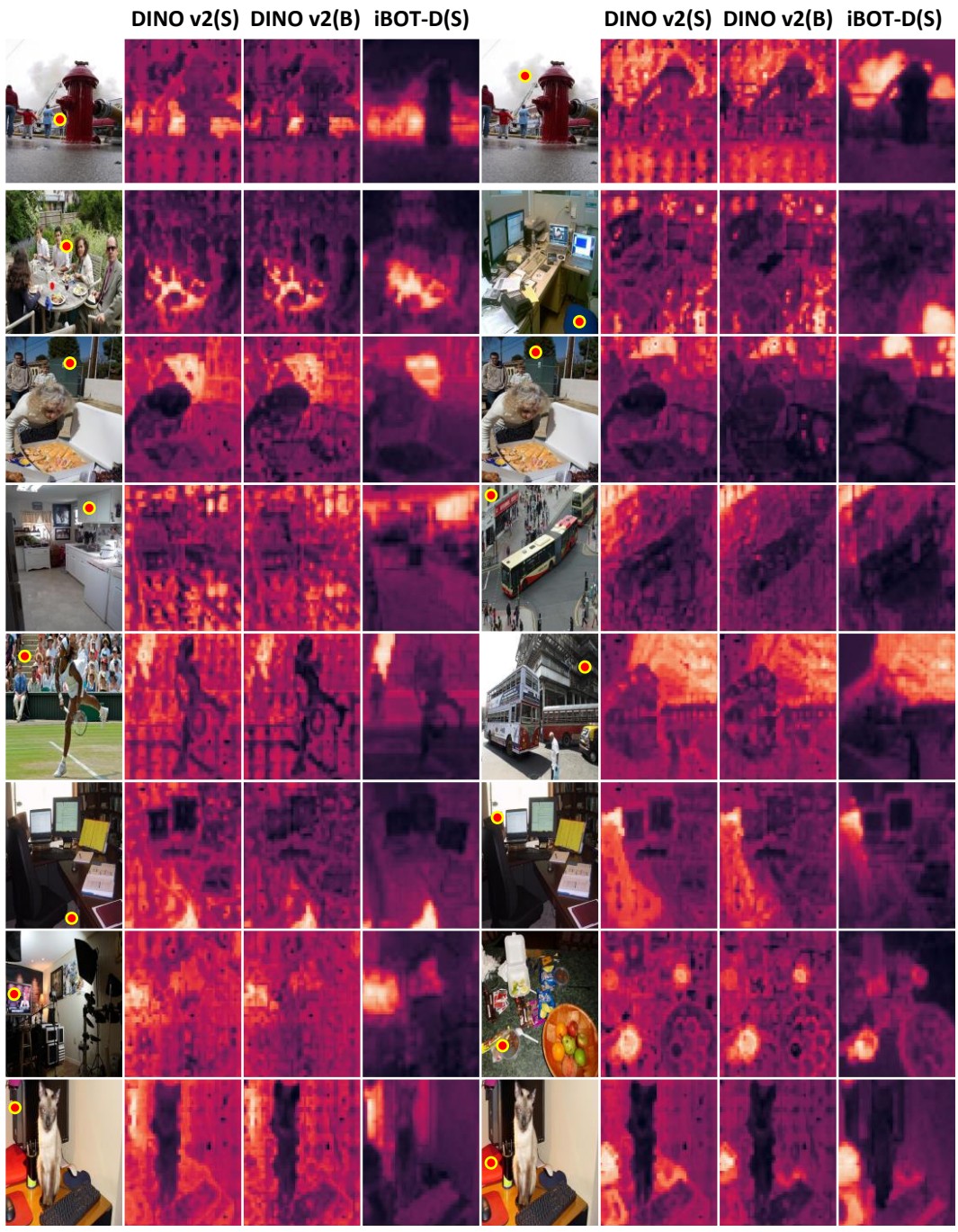

Figure 19: **Point affinity visualization on DINO v2 and iBOT-D.** The selected point-level feature is marked as a red dot. '(S)' denotes the backbone of ViT-S and 'B' denotes ViT-B.

