# OpenReview forum: "Mind Your Augmentation: The Key to Decoupling Dense Self-Supervised Learning"
_ICLR.cc/2024/Conference — ICLR 2024 poster_

### Official Review · Reviewer_VesV · 2023-10-30

**Soundness:** 3 good
**Presentation:** 2 fair
**Contribution:** 3 good
**Rating:** 6
**Confidence:** 3

**Summary:**

Submission 945 presents a visual self-supervision method suitable for dense (/local) tasks such as detection/segmentation, etc. It motivates itself by presenting demonstrations that show that attention maps (/point affinities) do not localize well to object-parts and this is due to the false positives generated by current self-supervision methods.

It presents a variant of CutMix for dense self-supervision that mixes tokens from the input image with tokens from an external image. The mixing strategy is developed such that a token from the input image is largely surrounded by tokens from the output image so that the input token is positionally “out of context”. It then presents a regularizer for any visual self-supervision loss such that features extracted from the “in context” and “out of context” tokens (taken from the input image) are similar.

**Strengths:**

- The presented work is reasonably thorough in its experiments and generality. In particular, I like that it presents a rationale for both ViTs and CNNs.
- While confusingly presented, the high-level idea of asking token-wise features to be invariant to some of their surrounding tokens is a simple idea and appears to lead to better attention maps and downstream performance.

**Weaknesses:**

(in no particular order)

### Poor presentation
Unfortunately, the unclear writing and figures significantly dampened any enthusiasm for this paper and it took multiple repeated readings to get a high-level sense of what is proposed. As a few examples,
- Paragraph 3 of the Introduction only makes sense if you already know the entire method.
- Figures 1(a) and 2(a) are really hard to understand and are used to motivate the work. For example, what do the red dashed lines, the arrows, and the “ViTs-based measuring pipeline” indicate in 2(a)?
- I don’t understand the written presentation of the mask generation (Sec. 4.1.1, par. 1) at all and its associated algorithm in the appendix uses undefined notation that is hard to follow (if it is defined elsewhere, please also add it to the caption). As a result, figure 4 does not immediately follow either.

IMO the paper requires a significant revision for clarity.

### Engineered token mixing has been done previously:
The proposed method has two main contributions: a cutmix style augmentation at the token level and a self-supervised loss leveraging that augmentation. However, while presented as new here, mixing tokens from different images in a carefully engineered way has been done before in TokenMix (ECCV’22, https://arxiv.org/abs/2207.08409 ) in the context of supervised classification and some other papers that follow up on it. Please discuss these works and clarify any differences, so as to better contextualize the key novel contribution of the self-supervised loss function that takes advantage of token mixing here.

### Unclear motivation and relationship to current work:
In my reading, the main motivation of this work is achieving higher quality part-level attention maps by reducing the dependence between a token’s features and its surroundings. It is then unclear to me if this strategy can then capture long-range nonlocal dependence – could you please comment on this point and clarify if I misunderstood?

Moreover, in recent work, DinoV2 (https://arxiv.org/abs/2304.07193) demonstrated that high-quality part-level representations can be learned by simply scaling up model and dataset sizes without considering the “coupling” between objects and background. I am not asking for comparisons, but I would like the response to briefly clarify the motivation of the proposed strategy in this context.

**Questions:**

### Suggestions:
- Please clarify the differences between TokenMix and similar token mixing works and this paper.
- Please revise and improve the writing and presentation of the first four sections of this paper to make it more immediately accessible.
- Please briefly discuss the motivation for the decoupling regularizer in the context of existing methods such as DinoV2 achieving high-quality part-level attention maps without considering the “coupling” phenomenon.
- Figure 5 on page 7 is what finally made the method click for me – please move it to earlier in the paper.

### Minor questions:
- Experimental clarifications would be beneficial: Why is 800 epochs of pretraining on COCO specifically chosen for all methods? How were the hyperparameters tuned for the baselines? Also, most of the experiments do not mention splits.
- Apart from the proposed masking, there appears to be no other augmentations mentioned and there is no code. Were standard augmentations such as jitters, flips, blur, etc. (https://github.com/bytedance/ibot/blob/main/main_ibot.py#L574) also used? Did the augmentation strategy match the existing methods on which decoupling was applied?

---

> ### Author Response · Authors · 2023-11-21
> **Response to Reviewer VesV**
>
> We sincerely thank Reviewer VesV for acknowledging our contributions. Below are detailed responses to the reviewer's concerns.
>
> >  **Q1: Presentation .**
>
> We appreciate the reviewer's feedback and acknowledge the concerns about clarity in the writing and figures. Thank you for bringing this to our attention. We rewrote some parts of the paper and also modified Figure 1(a) and Figure 2(a), We put Figure 4 right after the method as suggested by the reviewer, and made corresponding changes to make it easier to understand.
>
> >  **Q2: Differences between our work and current token mixing methods**
>
> Thank you for the valuable advice. We will add this to the related work. Although both methods utilize cutmix-style augmentation, our method differs significantly from them in both the mixing pipeline and loss design.
>
> Mixing Pipeline: Cutmix, TokenMix, and related works primarily apply rectangular or block-wise masks for mixing tokens, with a focus on image-level labels. In contrast, our Region Collaborative Cutout (RCC) generates background masks tailored for objects of varying scales. The ablation experiments in Section 6.1 demonstrate the superiority of RCC in de-coupling dense-level models compared to other mask types. Additionally, as highlighted in Section 6.3, RCC serves as a novel augmentation strategy beneficial for dense-level SSL pre-training.
>
> Loss Design: While cutmix-style methods in supervised learning aim to enhance the network's ability to capture long-range dependencies with the classification loss, our approach in dense-level SSL addresses a different concern. We empirically demonstrate in our experiments that the shared long-range context between tokens extracted by the student and teacher frameworks can potentially act as a shortcut in a self-supervised manner (without labels). To mitigate this, our de-coupling loss encourages the encoder to focus on capturing discriminative local semantics rather than collapsing to a single dominant long-range representation.
>
> >  **Q3.1: How to capture long-range nonlocal dependency ?**
>
> Thank you for the opportunity to clarify the main motivation of our work. Our primary goal is to reduce the dependence between a token's features and its surroundings, aiming to enhance the quality of part-level attention maps. The focus on decoupling dense-level representations is intended to encourage the network to prioritize local semantics over relying heavily on shared long-range context. This strategic approach aims to prevent different semantics from collapsing to the same representation, ensuring that each object remains discriminative. As evidence of the effectiveness of this strategy, we observe that methods incorporating the de-coupling loss outperform the original approach in effectively identifying objects across different locations.
>
>
> >  **Q3.2: Discussion about Dino V2 without "coupling"**
>
> Thanks for raising this question. Our method bears several differences with and advantages over DinoV2:
>
> Dataset Processing: Our motivation centers around training networks without extensive dataset curation, enabling self-supervised learning to be adaptable in various scenarios. Unlike DinoV2, which requires preprocessing the dataset and selecting images with single objects, our method aims to operate in a more versatile manner, not relying on specific dataset characteristics.
>
> Loss Function: DinoV2 employs the iBOT loss function, and we have demonstrated the compatibility of our method and its effectiveness in conjunction with the IBOT loss. This suggests that our approach can serve as a valuable tool to enhance the training of DinoV2 by addressing issues related to object coupling and improving overall understanding.
>
> We also conducted some visualization for better understanding, detailed comparisons are provided on the last page of the appendix. Despite DinoV2's excellent performance in identifying salient objects, we observed limitations in its ability to distinguish objects or scenes located in the background or less salient regions. The red point in the figure often exhibits high similarities with other regions from different semantic contexts. Increasing the capacity of the backbone network (from ViT-S to ViT-B) did not lead to improvements in DinoV2's performance in these scenarios. By contrast, our method with ViT-S demonstrates a better understanding of local semantics validating our initial intuition. Note that, DINOv2 also captures local semantics, and we are not aiming to compete with DINOv2 but rather proposing another potential avenue for advancing SSL.
>
> We hope that these clarifications highlight the distinctions and contributions of our method .
>
> >  **Minor questions: Hyper-parameters, datasets, and augmentations.**
> We will add those specifications to the paper, and of course, we will release our code very soon.
>
>
> We value the reviewer's input and will continue refining the content. We will update the paper to integrate the feedback.

---

> > ### Comment · Reviewer_VesV · 2023-11-23
> >
> > Thank you for your revision and the rebuttal. I will be maintaining my overall borderline accept rating, but raising the presentation score. Some minor assorted points:
> >
> >   - I agree that the text and figures are now clearer in parts. However, I still have some difficulty with key components. For example, the RCC masking procedure is hard to decipher from the description alone. I suggest that the authors eventually add an iterative figure somewhere in the paper that adds components step-by-step as this is the core contribution.
> >
> >   - I appreciate that there are new point affinity qualitative comparisons with DinoV2 (fig 18) where the proposed method achieves more interpretable maps. However, I do not think that it is fair to claim that the requisite data curation strategies for DinoV2 are hard to implement for new datasets. It only involves automated and scalable duplicate and near-duplicate removal (appendix A in their paper) which is straightforward to implement on new large datasets.
> >
> >   - (super minor point) On the note of DinoV2, in the final paper, it would be beneficial to briefly discuss [concurrent work](https://openreview.net/forum?id=2dnO3LLiJ1) that demonstrates a completely different approach to achieve interpretable attention maps without considering “coupling”. I am not at all suggesting a comparison or an in-depth discussion as this is a concurrent submission, but future readers may benefit from contrasting two approaches to the same goal in the final related work section.

---

> > > ### Author Response · Authors · 2023-11-23
> > > **Response to Reviewer VesV**
> > >
> > > We would like to thank the reviewer for your valuable time and constructive feedback. We will further improve the paper and incorporate your suggestions to improve the figures.
> > >
> > > It's important to clarify that we have no intention to compete with DINOv2. We acknowledge DINOv2's robustness and its simple yet impactful pre-processing approach. Our work tackles this problem from a different perspective,  as shown in our paper, our method can improve the iBOT loss, suggesting a supplementary enhancement on complex scenes for existing methods like DINOv2.
> > >
> > > We appreciate your mention of concurrent work without considering coupling. As mentioned in the concurrent paper (registers), the artifacts mainly encode global information instead of local semantics, we find it intriguing to delve into the connections between those artifacts and coupling issues in our future work. Additionally, we will see whether our method can be applied to reduce artifacts.

---

### Official Review · Reviewer_GFda · 2023-11-02

**Soundness:** 3 good
**Presentation:** 3 good
**Contribution:** 3 good
**Rating:** 8
**Confidence:** 2

**Summary:**

This paper presents a new method called Region Collaborative Cutout for self-supervised learning to alleviate the object coupling issue. This simple and straightforward method achieves evident gains over previous methods.

**Strengths:**

1. The paper is clearly written, and it is easy to catch the main motivation and solution. And I think the proposed method is well motivated.

2. Multiple previous methods are used as baselines to build the proposed Region Collaborative Cutout upon, and non-trivial improvements are observed. Besides, the method is proved effective for both CNNs and ViTs.

3. The ablation studies are comprehensive and convincing.

**Weaknesses:**

I am not very familiar with the research line of SSL. So I have no further suggestions for this paper. Generally, I like this simple yet effective method. It further highlights that constructing appropriate positive pairs by delicately designed strong augmentations is important.

However, one of my slight concerns is about the whole area of SSL since DINOv2 was released. It is pre-trained on extremely large-scale and curated data with several practical SSL optimization targets. It is very strong in many applications, such as retrieval, segmentation, and detection. Even with a frozen DINOv2 backbone, we can achieve state-of-the-art performance in some challenging tasks. Therefore, could the authors discuss the position of this submission by taking the recent SSL trend into consideration?

**Questions:**

No further questions.

---

> ### Author Response · Authors · 2023-11-21
> **Response to Reviewer GFda**
>
> >  **DINOv2 Discussion**
>
> We appreciate the reviewer's constructive suggestion about DINOv2. Our method bears several differences with and advantages over DINOv2:
>
> Dataset Processing: Our motivation centers around training networks without extensive dataset curation, enabling self-supervised learning to be adaptable in various scenarios. Unlike DinoV2, which requires preprocessing the dataset and selecting images with single objects, our method aims to operate in a more versatile manner, not relying on specific dataset characteristics.
>
> Loss Function: DinoV2 employs the iBOT loss function, and we have demonstrated the compatibility of our method and its effectiveness in conjunction with the IBOT loss. This suggests that our approach can serve as a valuable tool to enhance the training of DinoV2 by addressing issues related to object coupling and improving overall understanding.
>
> Experimental Observations: In our experiments, detailed comparisons are provided in the last page of the appendix. Despite DinoV2's excellent performance in identifying salient objects, we observed limitations in its ability to distinguish objects or scenes located in the background or less salient regions. The red point in the figure often exhibits high correlation with other regions from different semantic contexts. Increasing the capacity of the backbone network (from ViT-S to ViT-B) did not lead to improvements in DinoV2's performance in these scenarios. By contrast, our method with ViT-S demonstrates a better understanding of the entire scene, validating our initial intuition.

---

> > ### Comment · Reviewer_GFda · 2023-11-21
> >
> > Thank the authors for the feedback. Could the author provide more clarification on why DINOv2 is not good at distinguishing less salient regions? DINOv2 also considers the local features.

---

> > > ### Author Response · Authors · 2023-11-21
> > > **response to Reviewer GFda**
> > >
> > > Thank you for your prompt response. You are correct in noting that DINOv2 utilizes dense-level SSL loss, specifically the iBOT loss, allowing it to capture local features effectively.  DINOv2 enhances the performance by adopting a dedicated dataset curation strategy. This involves selecting images with only single objects and simple backgrounds to create a giant dataset, which is a different perspective from ours. We are not aiming to compete with DINOv2 but rather proposing another potential avenue for advancing SSL.  Meanwhile, since our loss is effective on iBOT loss to mitigate coupling issues, I hope our method can further improve DINOv2 since it still has some failure cases in complex scenes with multiple objects (as shown in Appendix H.3). If you have any further questions or need additional clarification, feel free to leave comments.

---

> > > > ### Comment · Reviewer_GFda · 2023-11-22
> > > >
> > > > Thank the authors for the response and clarification. I appreciate the contributions of this paper. Since my original score has already been high enough, I will keep the original score of Accept (8).

---

> > > > > ### Author Response · Authors · 2023-11-22
> > > > > **Response to Reviewer GFda**
> > > > >
> > > > > Thank you for your support. We appreciate your valuable time and constructive feedback.

---

### Official Review · Reviewer_wwZs · 2023-11-02

**Soundness:** 3 good
**Presentation:** 2 fair
**Contribution:** 3 good
**Rating:** 6
**Confidence:** 4

**Summary:**

This paper considers the problem of information leakage from the neighboring contextual regions in dense self-supervised learning. The contributions of this paper include the identification and confirmation of the coupling phenomenon when pairs have a limited overlap, the design of a decoupling branch, a novel region collaborative cutout augmentation, and the effectiveness of the proposed approach in the dense self-supervised learning frameworks. This approach can be applied to both CNNs and ViTs as the approach is only related to the augmentation.

**Strengths:**

This paper is well written and easy to follow.

The proposed method is simple and can be directly combined with existing dense self-supervised learning without introducing additional loss types.

The proposed method, RCC and decoupling branch, are demonstrated to be effective with several existing self-supervised learning methods in the experiments.

**Weaknesses:**

While the augmentation in Fig. 5 is easy to understand, the random masking in Fig. 2(a) is vague in terms of the illustration purpose.

The main text is expected to be self-contained. However, there are several cases where content is put in the appendix. For example, the figure 7(a) is useful to understand the definition of variables in equation 4, however, is put in the appendix. One possible way to address this problem is to merge Fig. 7(a) with part of Fig. 3. Again, the Alg 2 is another example for the understanding of the proposed RCC (region collaborative cutout).

The original contribution of the approach may be limited as the proposed RCC could be viewed as the combination of thresholding the cutout ratio within a region and filling the cutout region with background images. This is different from the Cutout, but more like combining existing strategies.

The order of Tables 4-6 does not match the appearance in the text.

**Questions:**

Context within the region. The shape of each object varies and is irregular. And the region is defined by a bounding box (at least in object detection task), There would be context information in the bounding box. How to measure or address the coupling or leakage for this part of information?

This method would introduce another hyperparameter, the threshold of cutout ratio. How to set this across different datasets?

Ablation study. The numbers about COCO Det. in Table 6 do not match the numbers reported in Table 2 for iBOT. This is because the training epochs are different. If we view the results in Table 2 as the converged one, the comparison made in Table 6 may not lead to a convincing conclusion. While three experiments are presented in this section, a more interesting ablation study would be the effectiveness of the decoupling branch.

While the authors claim that the proposed method can be combined with existing SSL methods, there is a concern that the proposed method may not work well for the method with contrastive loss within each batch. The reason for this is that the decoupling branch serves a similar purpose. However, there may be additional contribution as the losses are computed at different levels with different masks.

---

> ### Author Response · Authors · 2023-11-21
> **Response to Reviewer wwZs - 1**
>
> We sincerely thank Reviewer wwZs for acknowledging our contributions. Below are detailed responses to the reviewer's questions:
>
> >  **W1,W2,W4: Presentation.**
>
> 1Thanks for pointing out the unclear parts and for the constructive suggestions to improve the readability of our paper. Please check our updated version and let us know if we can still improve it.
> a. Figure 2: we have re-designed the figure based on your suggestion and updated it in the revised version.
> b. Figure 7 and Alg. 2. We have added Figure 7 in the context. We cannot find space for Alg. 2, but we have revised Section 4.1.1 to make it clearer.
> c. We have moved the table to their corresponding text.
>
> >  **W3: RCC is developed by combining existing strategies**
>
>
> We appreciate the reviewer's acknowledgment of the simplicity of our method. However, it is essential to note that our design is not merely a straightforward combination of existing strategies; rather, it involves a thoughtful and nontrivial adaptation to the unique challenges posed by dense SSL.
> The original Cutout algorithm is designed for CNN-based image-level augmentation methods; our RCC algorithm is specifically crafted to address the challenges of dense SSL and is applicable to both CNNs and Vision Transformers (ViTs). The essence of RCC lies in its grid-based bounding box generation, which introduces a collaborative cutout strategy to enhance feature robustness and mitigate the issues associated with coupling. Besides, Cutout aims to incorporate the background region, whereas RCC is used for storing the foreground.
>
> In our experiments, we conducted a thorough investigation of the compatibility of the original Cutout technique with our bounding box generation pipeline in the context of dense-level models. The combination did not yield favorable results and, in fact, led to performance degradation. We discussed this finding in Section 6.1 and provided detailed insights in Appendix G.1. Specifically, the experiments involving iBOT+Cutout and DenseCL+Cutout demonstrate degraded performance, with iBOT+Cutout ranking as the second-worst performer and DenseCL+Cutout even underperforming the original DenseCL. These results emphasize that addressing the issue of overcut is a critical factor in ensuring the effective collaboration of Cutout with dense-level SSL.
>
> >  **Q1: How to deal with objects with irregular shapes ?**
>
> Thanks for your question. This is one of the nice properties of RCC as it generates region-level bounding boxes at multiple scales and aspect ratios, allowing for a diverse scale of de-coupling. Consequently, the tokens representing objects of different sizes and shapes will be effectively encompassed by background image tokens, enabling de-coupling from irrelevant context at a finer grain. The intra-object coupling ratio in Sec 3.2 can measure the coupling issues in this context as it directly applies object segmentation masks instead of bounding boxes. The results in Fig. 3 and Section 6 show that de-coupling with RCC achieves the lowest CR-intra.
>
> >  **Q2: Cutout ratio in different Datasets.**
>
> The cutout ratio controls the balance between foreground regions for prediction and background regions for de-coupling. Throughout our extensive experiments, we have observed that this hyperparameter remains stable across different datasets. Its stability indicates that the proposed cutout ratio is robust and effective in achieving a favorable balance for both prediction and de-coupling, making it applicable to various datasets without the need for dataset-specific tuning. To provide further insight into the robustness of our approach, we conducted additional pre-training experiments with iBOT-D using different mask ratios on ImageNet-100[1] and OHMS (a subset of OpenImage) [2] for 200 epochs. The results consistently demonstrate the effectiveness of our de-coupling strategy across diverse datasets, reinforcing the notion that the cutout ratio parameter is a stable and generalizable setting.
>
> ##### 1. Pre-trained on ImageNet-100
> | **Model Name**|**Mask Ratio**|**COCO Det. AP**|
> | ----- | :-----: | :-----: |
> |iBOT|-|38.3|
> |iBOT-D|[0.2,0.4]|40.5|
> |iBOT-D |[0.3,0.5]|40.7|
> |iBOT-D |[0.4,0.6]|40.4|
>
> ##### 2. Pre-trained on OHMS
> | **Model Name**|**Mask Ratio**|**COCO Det. AP**|
> | ----- | :-----: | :-----: |
> |iBOT|-|41.2|
> |iBOT-D|[0.2,0.4]|43.8|
> |iBOT-D |[0.3,0.5]|44.2|
> |iBOT-D |[0.4,0.6]|44.0|

---

> > ### Comment · Reviewer_wwZs · 2023-11-21
> >
> > Thanks to the authors for the response.
> >
> > I want to follow up the Q1. Although multiple bounding boxes at multiple scales can be generated, to my understanding, the proposed method will only select only one out of them. That cause to the leakage of context information. Please correct me if I am wrong.

---

> > > ### Author Response · Authors · 2023-11-22
> > > **Response to your question**
> > >
> > > We are sending you an additional comment in case we misunderstand your question, and would like to provide more information about the issue of leakage, particularly concerning irregular shapes. In the case of using a specific rectangular bounding box as the foreground, the region outside of the irregular shape of objects will share the same context information as the key view, causing information leakage.  To tackle this problem, our RCC mask ensures that most query regions are encompassed by the background, which is sampled from a different image.  As demonstrated in Section 3, features of the surroundings can propagate to the foreground. Consequently, the features of the rectangular regions in the query view will be affected by the background, making it distinct from the corresponding regions in the key view. This intentional mismatch between key and query views mitigates the effects of information leakage, serving as a preventative measure against potential shortcuts in the learning process.
> > > Thanks to your insightful question, we will further investigate the use of masks generated by unsupervised segmentation methods to improve the results.
> > > Please feel free to share any further feedback or questions.

---

> ### Author Response · Authors · 2023-11-21
> **Response to Reviewer wwZs -2**
>
> >  **Q3-a: Ablation study in Tab. 6 about running epochs .**
>
> We acknowledge your concern regarding the duration of pre-training, and we appreciate the valuable insight. In our experiments, we observed consistent performance trends between 200-epoch and 800-epoch models when pre-trained with the same backbone and loss. However, due to limitations in computation, we primarily report the results of models pre-trained for 200 epochs in our study.
> To address the concern raised by the reviewer, we conducted additional experiments, pre-training iBOT-D with Blockwise Mask, the second-best model according to Table 6 (Table 4 in the updated version), for 800 epochs. The results clearly demonstrate that, even with an extended training schedule, our Region Collaborative Cutout (RCC) strategy continues to outperform the other mask strategies, supporting the robustness and effectiveness of our approach over longer pre-training durations. If the reviewer considers it necessary, we will continue running experiments for other ablation studies with 800 epochs.
>
> | **Model Name**|**COCO Det. AP**|
> | ----- | :-----: |
> |iBOT-D (RCC)| 45.1|
> |iBOT-D (Blockwise)| 44.0|
>
> >  **Q4: Compatibility of our method with methods employing contrastive learning.**
>
> DenseCL indeed employs contrastive learning, and our proposed method still integrates effectively within such a framework. In our approach, each decoupled view samples only one image as the background, and the decoupling loss is applied exclusively to the foreground patches, minimizing the impact of intra-batch contrast. To address the reviewer's concerns, we performed additional experiments where we randomly sampled the background image from the entire dataset outside the batch at each iteration. The results from these experiments, with models pre-trained on COCO for 200 epochs, demonstrate that the choice of sampling background images within or outside the batch has a negligible impact on the model's performance.
>
> | **Model Name**|**COCO Det. AP**|
> | ----- | :-----: |
> |iBOT-D (Within-Batch)| 42.0 |
> |iBOT-D (Out-of-Batch)| 42.2 |
>
>
> We appreciate the insightful questions and feedback. We plan to incorporate the provided suggestions into our paper for improvement. If you have any additional inquiries or require further clarification, please feel free to reach out. Thank you for your valuable input.
>
>
> __References__
>
> [1] Tian, Yonglong, Dilip Krishnan, and Phillip Isola. "Contrastive multiview coding." Computer Vision–ECCV 2020: 16th European Conference, Glasgow, UK, August 23–28, 2020, Proceedings, Part XI 16. Springer International Publishing, 2020.
>
> [2] Mishra, Shlok, et al. "Object-aware cropping for self-supervised learning." arXiv preprint arXiv:2112.00319 (2021).

---

> ### Author Response · Authors · 2023-11-21
> **response to Reviewer wwZs**
>
> Thank you for your prompt response. Our RCC generates cutout masks for each bounding box, and each image contains N^2 bounding boxes, i.e. N^2 cutout regions in total. Note that the size and position of bounding boxes in each image differ, thus the masks generated by RCC also vary. Let me clarify the intuition behind our RCC. Existing augmentations, such as cutout, are all image-level. Once the cutout region is large enough to cause incorrect alignment between the teacher and student encoder, the feature of the nearby region will be pushed towards the masked region, causing coupling. Therefore, as one of our main modifications, we iteratively restore the overcut regions in each bounding box and perform the cutout. Our primary goal is to achieve effective augmentation at a dense level without causing inaccurate alignment. We present some examples of the RCC masks in Fig.12, and the pseudo code is available in Alg. 2 in our paper. Please feel free to provide any feedback.

---

### Meta-Review · Area_Chair_TXN3 · 2023-12-05

**Metareview:**

This paper presents a way for self-supervised learning tackling dense visual prediction tasks such as segmentation and detection. The authors propose a solution inspired by CutMix augmentation that combines visual signals from two images to create in context and out of context visual tokens. The learning signal consists of preserving the similarity of the features from both types of tokens.
The approach provides strong gains on benchmark tasks and works with both ViT and CNN architectures.
Like many of the reviewers, the AC also found the presentation of this work to be confusing and not self-contained.

**Justification For Why Not Higher Score:**

The presentation of this paper and its relation to existing works can be improved. This point has been raised by multiple reviewers and does limit the impact of this work.

**Justification For Why Not Lower Score:**

The paper has done a good job of demonstrating that there is a coupling problem present in current ssl works. The proposed solution improves performance for multiple architectures and is simple. This will likely benefit future research in dense SSL.

---

### Decision · Program_Chairs · 2024-01-16

Accept (poster)